# Sequential Decision Making with Expert Demonstrations under Unobserved Heterogeneity

**Vahid Balazadeh**
University of Toronto
vahid@cs.toronto.edu

**Keertana Chidambaram**
Stanford University
vck@stanford.edu

**Viet Nguyen**
University of Toronto
viet@cs.toronto.edu

**Rahul G. Krishnan**[*]
University of Toronto
rahulgk@cs.toronto.edu

**Vasilis Syrgkanis**[*]
Stanford University
vsyrgk@stanford.edu

## Abstract

We study the problem of online sequential decision-making given auxiliary demonstrations from *experts* who made their decisions based on unobserved contextual information. These demonstrations can be viewed as solving related but slightly different problems than what the learner faces. This setting arises in many application domains, such as self-driving cars, healthcare, and finance, where expert demonstrations are made using contextual information, which is not recorded in the data available to the learning agent. We model the problem as zero-shot meta-reinforcement learning with an unknown distribution over the unobserved contextual variables and a Bayesian regret minimization objective, where the unobserved variables are encoded as parameters with an unknown prior. We propose the Experts-as-Priors algorithm (ExPerior), an empirical Bayes approach that utilizes expert data to establish an informative prior distribution over the learner's decision-making problem. This prior distribution enables the application of any Bayesian approach for online decision-making, such as posterior sampling. We demonstrate that our strategy surpasses existing behaviour cloning, online, and online-offline baselines for multi-armed bandits, Markov decision processes (MDPs), and partially observable MDPs, showcasing the broad reach and utility of ExPerior in using expert demonstrations across different decision-making setups.

## 1 Introduction

Reinforcement learning (RL) has found success in complex decision-making tasks, spanning areas such as game playing [1, 2, 3], robotics [4, 5], and aligning with human preferences [6]. However, RL's considerable sample inefficiency, necessitating millions of training frames for convergence, remains a significant challenge. A notable body of work within RL has been dedicated to integrating expert demonstrations to accelerate the learning process, employing strategies like offline pretraining [7] and the use of combined offline-online datasets [8, 9]. While these approaches are theoretically sound and empirically validated [10, 11], they typically presume homogeneity between the offline and online datasets. A vital question arises about the effectiveness of these methods when expert data embody heterogeneous tasks, indistinguishable by the learner.

An important example of such heterogeneity is in situations where experts operate with additional information not available to the learner, a scenario previously explored in imitation learning with unobserved contexts [12, 13, 14, 15]. Existing literature either relies on the availability of experts to query during training [16, 17, 18, 19] or focuses on the assumptions that enable imitation learn-

---

[*]Equal contribution. Our code is accessible at https://github.com/vdblm/experior

38th Conference on Neural Information Processing Systems (NeurIPS 2024).

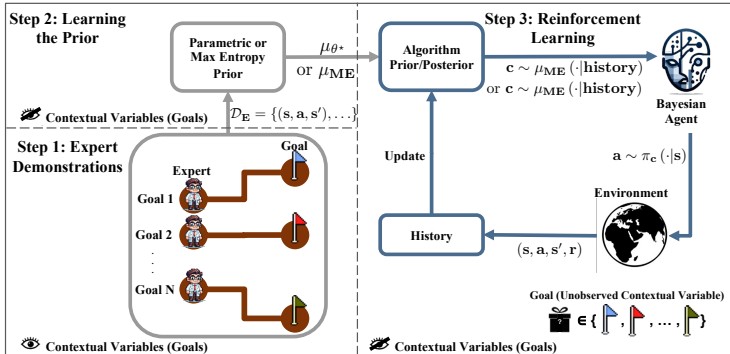

Figure 1: Illustration of ExPerior in a goal-oriented task. Step 1 (Offline): The experts demonstrate their policies for different goal types while observing them. Step 2 (Offline): The expert data $\mathcal{D}_E$ only contains the trajectories states/actions — goal types are not collected. We form an informative prior distribution over the goal types (unobserved factors) using $\mathcal{D}_E$. Step 3 (Online): The goal type is unknown but drawn from the same distribution of goals in Step 1. The learner uses the learned prior for posterior sampling.

ing with unobserved contexts, sidestepping online reward-based interactions [20, 21]. Recent contributions by Hao et al. [22, 23] suggest using offline expert data for online RL, albeit without accounting for unobserved variations. Our work addresses the more general challenge of online decision-making given auxiliary offline expert data with *unobserved* heterogeneity. We view such demonstrations as solving related yet distinct problems from those faced by the learner, where differences remain invisible to the learner. For instance, in a personalized education scenario, while a learning agent can observe characteristics like grades or demographics, it might remain oblivious to factors such as learning styles, which are visible to an expert teacher and can significantly influence teaching methods. Naïve imitation without access to this "private" information will only learn a single policy for each observed characteristic [24], leading to sub-optimal actions. On the other hand, a purely online approach requires extensive trial and error to result in meaningful decisions.

We integrate offline expert data with online RL, treating the scenario as a zero-shot meta-reinforcement learning (meta-RL) problem with an unknown distribution over unobserved contextual variables. Unlike typical meta-RL frameworks where the learner is exposed to multiple instances during training (different students in our example) to learn the underlying distribution [25, 26], our approach only leverages offline expert data to infer the distribution of unobserved factors, embodying a *zero-shot* meta-RL paradigm [27].

**Contributions.** We define a Bayesian regret minimization objective and consider unobserved variables as parameters under an unknown prior distribution. We use empirical Bayes to derive an informative prior over the unobserved variables from expert data. We use the learned prior distribution to drive exploration in the online RL task, using approaches like posterior sampling [28]. We propose two procedures to learn such a prior: (1) a parametric approach that can utilize any existing knowledge about the parametric form of the prior distribution, and (2) a nonparametric approach that employs the principle of maximum entropy when such prior knowledge does not exist. We call our framework Experts-as-Priors or ExPerior for short. See Figure 1 for a goal-oriented RL example. ExPerior outperforms existing offline, online, and offline-online baselines in multi-armed bandits, Markov decision processes (MDPs), and partially observable MDPs. For multi-armed bandits, we find the Bayesian regret incurred by ExPerior is proportional to the entropy of the optimal action under the prior distribution, aligning with the entropy of expert policy if the experts are optimal. We introduce a frequentist algorithm for multi-armed bandits and prove a Bayesian regret bound proportional to a term that closely resembles the entropy of the optimal action. Our results suggest using the entropy of expert demonstrations to evaluate the impact of unobserved factors.

## 2 Related Work

Our work is an addition to the recent body of reinforcement learning research that leverages offline demonstrations to speed up online learning [29, 10, 30, 7, 9]. Classic algorithms such as DDPGfD [31] and DQfD [32] achieve this by combining imitation learning and RL. They modify DDPG [5] and DQN [1] by warm-starting the algorithms' replay buffers with expert trajectories and ensuring that the offline data never gets overridden by online trajectories. Closely related to our study is the meta-RL literature, which aims to accelerate learning in a given RL task by using

prior experience from related tasks [33, 34, 35]. These papers present model-agnostic meta-learning training objectives to maximize the expected reward from novel tasks as efficiently as possible.

Two unique features distinguish our problem from the settings considered above. First, our setting assumes heterogeneity within the offline data and with the online RL task that is unobserved to the learner, while the (optimal) experts are privy to that heterogeneity. Second, we assume the learner will only interact with one online task, making our setup similar to zero-shot meta-RL [27, 36, 37]. Most similar to our work is the ExPLORe algorithm [38], which assigns optimistic rewards to the offline data during the online interaction and runs an off-policy algorithm using both online and labelled offline data as buffers. For our setting, the algorithm incentivizes the learner to explore the expert trajectories, leading to faster convergence. We consider this work one of our baselines.

Our methodology utilizes only the state-action trajectory data from expert demonstrations without task-specific information or reward labels. Other similar methods require additional offline information. For example, Nair et al. [30] assume that the offline data contains the reward labels and use that to pre-train a policy, which is then fine-tuned online. Mendonca et al. [39] require task labelling for each trajectory and use the offline data to learn a single meta-learner. Similarly, Zhou et al. [40] and Rakelly et al. [41] require the task label and reward labels. They then infer the task during online interaction and use the task-specific offline data. Lee et al. [42] requires a large amount of noisy expert data with reward labels, in addition to the optimal trajectory data, for good performance. Finally, our methodology builds on posterior sampling [43]. Hao et al. [22, 23] consider a similar problem using posterior sampling to leverage offline expert demonstration data to improve online RL. However, they assume homogeneity between the expert data and online tasks. In contrast, our setting accounts for heterogeneity.

## 3 Problem Setup

**Decision Model for Unobserved Heterogeneity.** To account for unobserved heterogeneity, we consider a generalization of finite-horizon Markov Decision Processes (MDPs) with a notion of probabilistic contextual variables [44, 13, 21]. The underlying model for the MDP will additionally depend on an unobserved variable that encapsulates the information hidden from the learner. For example, consider a personalized education setup where teaching a student corresponds to a task, and the learning agent can observe students' characteristics, like their demographic status and grades. Other factors, such as the student's learning style (e.g., visual learners or self-study), may not be readily available, even though they are important in determining the optimal teaching style.

Let $\mathcal{C}$ be the set of all *unobserved* context variables that can describe the unobserved heterogeneity of the decision-making problem (e.g., the set of all possible learning styles). A contextual MDP $\mathcal{M} = (\mathcal{S}, \mathcal{A}, \mathcal{T}, R, H, \rho, \mu^\star)$ is parameterized by states $\mathcal{S}$, actions $\mathcal{A}$, transition function $\mathcal{T} : \mathcal{S} \times \mathcal{A} \times \mathcal{C} \to \Delta(\mathcal{S})$, reward function $R : \mathcal{S} \times \mathcal{A} \times \mathcal{C} \to \Delta(\mathbb{R})$, horizon $H > 0$, initial state distribution $\rho \in \Delta(\mathcal{S})$, and context distribution $\mu^\star$. We assume the transition/reward functions and $\mu^\star$ are unknown, and for simplicity, $\rho$ does not depend on the context variable. For each unobserved context $c \sim \mu^\star$, we consider $T$ episodes, where at the beginning of each episode $t \in [T]$, an initial state $s_1 \sim \rho$ is sampled. Then, at each timestep $h \in [H]$, the learner chooses an action $a_h \in \mathcal{A}$, observes a reward $r_h \sim R(s_h, a_h, c)$ and the next state $s_{h+1} \sim \mathcal{T}(s_h, a_h, c)$. Without loss of generality, we assume the states are partitioned by $[H]$ to make the notation invariant to the timestep. Let $\Pi$ be the set of all Markovian policies. For a policy function $\pi : \mathcal{S} \to \Delta(\mathcal{A}) \in \Pi$ and context variable $c$, we define the value function $V_c(\pi) = \mathbb{E}\left[\sum_{h=1}^{H} r_h \mid \pi, c\right]$ and the Q-function as $Q_c^\pi(s, a) := \mathbb{E}\left[\sum_{h'=h}^{H} r_{h'} \mid s_h = s, a_h = a, \pi, c\right]$ for all $s \in \mathcal{S}, a \in \mathcal{A}$. Moreover, we define the optimal policy for a context variable $c \in \mathcal{C}$ as $\pi_c := \arg\max_{\pi \in \Pi} V_c(\pi)$. Note that since the context variable is unobserved, the learner's policy will not depend on it. The learning agent's goal is to learn history-dependent distributions $p^1, \ldots, p^T \in \Delta(\Pi)$ over Markovian policies to minimize the expected regret, defined as $\text{Reg} := \mathbb{E}_{c \sim \mu^\star}\left[\sum_{t=1}^{T} V_c(\pi_c) - \mathbb{E}_{\pi^t \sim p^t}[V_c(\pi^t)]\right]$.

In the personalized education example, the above setup assumes a fixed distribution $\mu^\star$ over the set of learning styles and aims to minimize expected regret over the population of students. Our setup and regret assume the unobserved factors remain fixed during training. This captures scenarios wherein the unobserved variables correspond to less-variant factors (a student's learning style is more likely to remain unchanged). No learning algorithm can control the regret value if we allow the unobserved factors to change arbitrarily throughout $T$ episodes without access to hidden information; consider

a two-armed bandit with a context value drawn with uniform probability from $\mathcal{C} = \{c_1, c_2\}$ that can change at each episode. Assume the expected reward of the first arm under $c_1$ and $c_2$ is one and zero, respectively, and it is reversed for the other arm. Any algorithm that does not have access to $c$ would result in linear regret since each action is sub-optimal with a probability of $0.5$, independent of the algorithm's choice.

**Remark.** Our setup can be formulated as a Bayesian model parameterized by $\mathcal{C}$, and our regret can be seen as the Bayesian regret of the learner. However, the distribution $\mu^\star$ is not the learner's prior belief about the true model as it is often formulated in Bayesian learning, but a distribution over potential contexts that the learner can encounter. Our setup can thus be seen as a meta-learning problem. In fact, it is *zero-shot* meta-learning since we do not assume having access to more than one online RL task during training — we only learn the prior distribution using the offline data.

**Expert Demonstrations.** In addition to the online setting described above, we assume the learner has access to an offline dataset of expert demonstrations $\mathcal{D}_E$, where each demonstration $\tau_E = (s_1, a_1, s_2, a_2, \ldots, s_H, a_H, s_{H+1})$ refers to an interaction of the expert with a decision-making task during a *single* episode, containing the actions made by the expert and the resulting states. We assume that the unobserved context variables for $\mathcal{D}_E$ are drawn i.i.d. from distribution $\mu^\star$, and the expert had access to such unobserved variables (private information) during their decision-making. Moreover, we assume the expert follows a near-optimal strategy [22, 23].

**Assumption 1** (Noisily Rational Expert). For any $c \in \mathcal{C}$, experts select actions based on a distribution defined as $p_E(a \mid s\,;\,c) \propto \exp\{\beta \cdot Q_c^{\pi_c}(s, a)\}$, for all $s \in \mathcal{S}, a \in \mathcal{A}$, and some known competence value of $\beta \in [0, \infty]$. In particular, the expert follows the optimal policy if $\beta \to \infty$.

We assume experts do not provide any rationale for their strategy, nor do we have access to rewards in the offline data; this is a combination of imitation and online learning rather than offline RL [22].

## 4 Experts-as-Priors Framework for Unobserved Heterogeneity

Our goal is to leverage offline data to help guide the learner through its interaction with the decision-making task. The key idea is to use expert demonstrations to infer a *prior* distribution over $\mathcal{C}$ and then to use a Bayesian approach such as posterior sampling [28] to utilize the inferred prior for a more informative exploration. If the current context is from the same distribution of contexts in the offline data, we expect that using such priors will lead to faster convergence to optimal trajectories compared to the commonly used non-informative priors. Consider the personalized education example. Suppose we have gathered offline data on an expert's teaching strategies for students with similar observed information like grade, age, location, etc. The teacher can observe more fine-grained information about the students that is generally absent from the collected data (e.g., their learning style). Our work relies on the following observation: The space of the optimal strategies for students with similar observed information but different learning styles is often much smaller than the space of all possible strategies. With the inferred prior distribution, the learner needs only to focus on the span of potentially optimal strategies for a new student, allowing for significantly more efficient exploration.

We resort to empirical Bayes and use maximum marginal likelihood estimation [45] to construct a prior distribution from $\mathcal{D}_E$. Given a probability distribution (prior) $\mu$ on $\mathcal{C}$, the marginal likelihood of an expert demonstration $\tau_E = (s_1, a_1, s_2, a_2, \ldots, s_H, a_H, s_{H+1}) \in \mathcal{D}_E$ is given by

$$P_E(\tau_E\,;\,\mu) = \mathbb{E}_{c \sim \mu}\left[\rho(s_1) \cdot \prod_{h=1}^{H} p_E(a_h \mid s_h\,;\,c)\,\mathcal{T}(s_{h+1} \mid s_h, a_h, c)\right]. \qquad (1)$$

We aim to find a prior distribution to maximize the log-likelihood of $\mathcal{D}_E$ under the model described in (1). This is equivalent to minimizing the KL divergence between the marginal likelihood $P_E$ and the empirical distribution of expert demonstrations, which we denote by $\widehat{P}_E$. In particular, we form an uncertainty set over the set of plausible priors as $\mathcal{P}(\epsilon) := \left\{\mu\,;\,D_{KL}\left(\widehat{P}_E \,\big\|\, P_E(\cdot\,;\,\mu)\right) \leq \epsilon\right\}$, where the value of $\epsilon$ can be chosen based on the number of samples so the uncertainty set contains the true prior with high probability [27]. However, the set of plausible priors does not uniquely identify the appropriate prior. In fact, even for $\epsilon = 0$, $\mathcal{P}(\epsilon)$ can have infinite plausible priors. To solve this ill-posed problem, we propose two approaches, parametric and nonparametric prior learning.

**Parametric Experts-as-Priors.** For settings where we have existing knowledge about the parametric form of the prior, we can directly apply maximum marginal likelihood estimation to learn it. In

particular, we define the parametric expert prior as the following. Note that we can calculate the gradients of the marginal likelihood using the score function estimator [46].

**Definition 1** (Parametric Expert Prior). *Let $\Theta$ be a set of plausible prior distribution parameters (e.g., Beta distribution parameters for a Bernoulli bandit). We call $\mu_{\theta^\star}$ a parametric expert prior, iff $\theta^\star \in \arg\min_{\theta \in \Theta} \sum_{\tau \in \mathcal{D}_E} -\log \mathrm{P_E}(\tau\,;\,\mu_\theta)$.*

**Nonparametric Experts-as-Priors.** For settings where there is no existing knowledge on the parametric form of the prior, we can employ the principle of maximum entropy to choose the *least informative* prior that is compatible with expert data:

**Definition 2** (Max-Entropy Expert Prior). *Let $\mu_0$ be a non-informative prior on $\mathcal{C}$ (e.g., a uniform distribution). Given some $\epsilon > 0$, we define the maximum entropy expert prior $\mu_{\mathrm{ME}}$ as the solution to the following optimization problem:*

$$\mu_{\mathrm{ME}} = \arg\min_\mu \ \mathrm{D_{KL}}(\mu \,\|\, \mu_0) \quad \text{s.t.} \quad \mu \in \mathcal{P}(\epsilon). \tag{2}$$

Note that the set of plausible priors $\mathcal{P}(\epsilon)$ is a convex set, and therefore, (2) is a convex optimization problem. We can derive the solution to problem (2) using Fenchel's duality theorem [47, 48]:

**Proposition 1** (Max-Entropy Expert Prior). *Let $N = |\mathcal{D}_E|$ be the number of demonstrations in $\mathcal{D}_E$. For each $c \in \mathcal{C}$ and demonstration $\tau_E = (s_1, a_1, s_2, a_2, \ldots, s_H, a_H, s_{H+1}) \in \mathcal{D}_E$, define $m_{\tau_E}(c)$ as the (partial) likelihood of $\tau_E$ under $c$, i.e., $m_{\tau_E}(c) = \prod_{h=1}^{H} p_E(a_h \mid s_h\,;\,c)\,\mathcal{T}(s_{h+1} \mid s_h, a_h, c)$.*

*Denote $\mathbf{m}(c) \in \mathbb{R}^N$ as the vector with elements $m_{\tau_E}(c)$ for $\tau_E \in \mathcal{D}_E$. Moreover, let $\lambda^\star \in \mathbb{R}^{\geq 0}$ be the optimal solution to the Lagrange dual problem of (2). Then, the solution to optimization (2) is:*

$$\mu_{ME}(c) = \lim_{n \to \infty} \frac{\exp\left\{\mathbf{m}(c)^\top \boldsymbol{\alpha}_n\right\}}{\mathbb{E}_{c' \sim \mu_0}\left[\exp\left\{\mathbf{m}(c')^\top \boldsymbol{\alpha}_n\right\}\right]},$$

*where $\{\boldsymbol{\alpha}_n\}_{n='1}^{\infty}$ is a sequence converging to the following supremum:*

$$\sup_{\boldsymbol{\alpha} \in \mathbb{R}^N} \ -\log \mathbb{E}_{c \sim \mu_0}\left[\exp\left\{\mathbf{m}(c)^\top \boldsymbol{\alpha}\right\}\right] + \frac{\lambda^\star}{N} \sum_{i=1}^{N} \log\left(\frac{N \cdot \alpha_i}{\lambda^\star}\right). \tag{3}$$

The proof is provided in Appendix A.3. Instead of solving for $\lambda^\star$, we set it as a hyperparameter and then solve (3). Even though Proposition 1 requires the correct form of Q-functions for different values of $c$, we will see in the following sections that we can parameterize the Q-functions and treat those parameters as a proxy for the unobserved factors. Once such a prior is derived, we can employ any Bayesian approach for decision-making. We provide a pseudo-algorithm for ExPerior in Algorithm 1. The following sections will detail the algorithm for bandits and MDPs.

---

**Algorithm 1** Experts-as-Priors (ExPerior-MaxEnt)

---

1: **Input:** Expert demonstrations $\mathcal{D}_E$, Reference distribution $\mu_0$, $\lambda^\star \geq 0$, and unknown $c \sim \mu^\star$.
2: $\mu_{\mathrm{ME}} \leftarrow \textsc{MaxEntropyExpertPrior}(\mu_0, \mathcal{D}_E, \lambda^\star)$
3: *history* $\leftarrow \{\}$
4: **for** episode $t \leftarrow 1, 2, \ldots$ **do**
5:      sample $c_t \sim \mu_{\mathrm{ME}}(\cdot \mid \textit{history})$          // posterior sampling
6:      **for** timestep $h \leftarrow 1, 2, \ldots, H$ **do**
7:          take action $a_h^t \sim \pi_{c_t}(\cdot \mid s_h)$
8:          observe $r_h^t \sim R(s_h^t, a_h^t, c)$, $s_{h+1}^t \sim \mathcal{T}(s_h^t, a_h^t, c)$, and append $(a_h^t, r_h^t, s_{h+1}^t)$ to *history*
9:      **end for**
10: **end for**

---

## 5 Learning in Bandits

$K$**-armed Bandits.** For $K$-armed bandits, note that $\mathcal{S} = \emptyset$, $H = 1$, and $\mathcal{A} = \{1, \ldots, K\}$. Each expert demonstration $\tau_E = a$ will be the pulled arm by the expert for a particular bandit, and the (partial) likelihood function in Proposition 1 can be simplified as $m_{\tau_E}(c) = p_E(a\,;\,c)$. This likelihood function only depends on the context variable $c$ through the expert policy $p_E$, and since $p_E$ only depends on $c$ through the mean reward function (Assumption 1), we can consider the set of mean reward functions as a proxy for the unobserved context variables $\mathcal{C}$. e.g. in a Bernoulli $K$-armed bandit setting, we can define $\mathcal{C}_{\mathrm{Ber}} = \left\{a \mapsto \langle \mathbf{e}_a, \vartheta \rangle\,;\,\vartheta \in [0, 1]^K\right\}$.

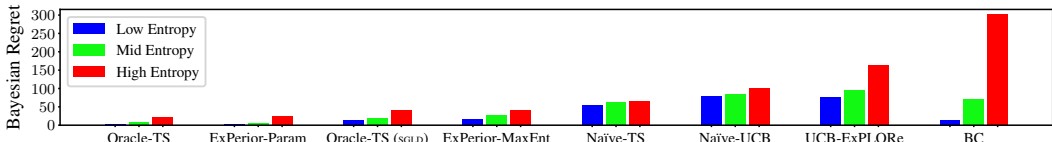

Figure 2: The Bayesian regret of ExPerior and baselines for $K$-armed Bernoulli bandits ($K = 10$). We consider three categories of prior distributions based on the entropy of the optimal action.

**Posterior Sampling.** With the above parameterizations of $\mathcal{C}$, we can use Proposition 1 to derive the maximum entropy prior distribution over the context parameters. However, we cannot sample from the exact posterior since the derived prior is not a conjugate prior for standard likelihood functions. Instead, we resort to approximate posterior sampling via stochastic gradient Langevin dynamics (SGLD) [49]. We call this method `ExPerior-MaxEnt` in our experiments. We also employ a parametric approach as discussed in section 4, which we call `ExPerior-Param`. In particular, we use the Beta distribution as our prior model and learn the parametric expert prior in Definition 1.

In the following, we evaluate our approach compared to online methods that do not use expert data and offline behaviour cloning. We provide an empirical regret analysis for ExPerior based on the informativeness of expert data, number of actions, and number of training episodes. We also discuss the robustness of ExPerior to misspecified expert models and the advantage of `ExPerior-MaxEnt` to `ExPerior-Param` when the parametric prior model is misspecified. To characterize the effect of expert data on the learner's performance, we propose an alternative for $K$-armed bandits inspired by the successive elimination and derive a Bayesian regret bound for it.

**Experiments.** We consider $K$-armed Bernoulli bandits for our experimental setup. We evaluate the learning algorithms in terms of the Bayesian regret over multiple (prior) distributions $\mu^\star$ over the unobserved contexts. In particular, we consider up to $N_{\mu^\star} = 64$ different beta distributions, where their parameters are chosen to span a different range of heterogeneity, consisting of tasks with various expert data informativeness. To estimate the Bayesian regret, we sample $N_{\text{task}} = 128$ bandit tasks from each prior distribution and calculate the average regret. We use $N_E = 1000$ expert demonstrations for each prior distribution in our experiments. We compare ExPerior to the following baselines: (1) Behaviour cloning (`BC`), which learns a policy by minimizing the cross-entropy loss between the expert demonstrations and the agent's policy solely based on offline data. (2) Naïve Thompson sampling (`Naïve-TS`) that chooses the action with the highest sampled mean from a posterior distribution under an uninformative prior. (3) Naïve upper confidence bound (`Naïve-UCB`) algorithm that selects the action with the highest upper confidence bound. Both `Naïve-TS` and `Naïve-UCB` ignore expert demonstrations. (4) `UCB-ExPLORe`, a variant of the algorithm proposed by Li et al. [38] tailored to bandits. It labels the expert data with optimistic rewards and then uses it alongside online data to compute the upper confidence bounds for exploration, and (5) `Oracle-TS`, which performs exact Thompson sampling with the true prior distribution $\mu^\star$. For a fair comparison, we also consider a variant of `Oracle-TS`, which uses SGLD for approximate posterior sampling.

**Comparison to baselines.** Figure 2 demonstrates the average Bayesian regret for various prior distributions over $T = 1,500$ episodes with $K = 10$ arms. To better understand the effect of expert data, we categorize the prior distributions by the entropy of their optimal actions into low entropy (less than 0.8), high entropy (greater than 1.6), and medium entropy. `Oracle-TS` and `ExPerior-Param` outperform other baselines, yet the performance of `ExPerior-MaxEnt` is comparable to the SGLD variant of `Oracle-TS`. This indicates that the maximum entropy prior derived from Proposition 1 closely approximates the true prior distribution, $\mu^\star$, and the performance difference with `Oracle-TS` is primarily due to approximate posterior sampling. Moreover, the pure online algorithms `Naïve-TS` and `Naïve-UCB`, which disregard expert data, display similar performance across different entropy levels, contrasting with other algorithms that show significantly reduced regret in low-entropy contexts. This underlines the impact of expert data in settings where the unobserved confounding has less effect on the optimal actions. Specifically, in the extreme case of no unobserved heterogeneity, `BC` is anticipated to yield optimal performance. Additionally, `Naïve-UCB` surpasses `UCB-ExPLORe` in medium and high entropy settings, possibly due to the over-optimism of the reward labelling in Li et al. [38], which can hurt the performance when the expert demonstrations are uninformative.

**Empirical regret analysis for Experts-as-Priors.** We examine how the quality of expert demonstrations affects the Bayesian regret achieved by ExPerior. Settings with highly informative demon-

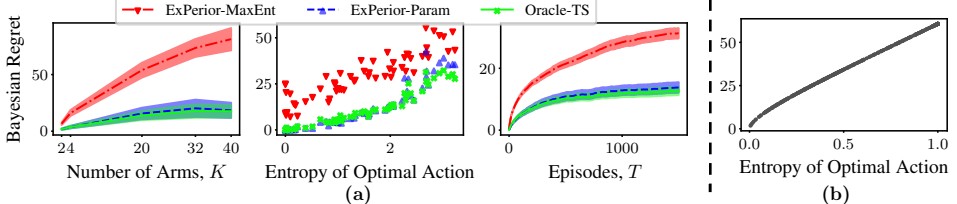

Figure 3: (a) Empirical analysis of ExPerior's regret in Bernoulli bandits based on the (left) number of arms, (middle) entropy of the optimal action, and (right) number of episodes. (b) The regret bound from Theorem 2 vs. the entropy of the optimal action. The linear relationship is consistent with the middle panel of (a).

Table 1: Ablation experiments to assess the robustness of ExPerior to misspecified expert models. Random-optimal experts choose the optimal action with probability $\gamma$ and choose random actions with probability $1 - \gamma$. `ExPerior-MaxEnt` achieves consistent out-performance by setting the hyperparameter $\beta = 10$. while `ExPerior-Param` get almost similar results for $\beta = 1$ and $\beta = 2.5$.

| | Optimal | Noisily-Rational | | | | Random-Optimal | | | |
|---|---|---|---|---|---|---|---|---|---|
| | | $\beta = 0.1$ | $\beta = 1$ | $\beta = 2.5$ | $\beta = 10$ | $\gamma = 0.0$ | $\gamma = 0.25$ | $\gamma = 0.5$ | $\gamma = 0.75$ |
| ExPerior-MaxEnt ($\beta = 0.1$) | $51.7 \pm 5.1$ | $52.3 \pm 5.3$ | $52.3 \pm 5.3$ | $52.0 \pm 5.1$ | $51.7 \pm 5.0$ | $52.3 \pm 5.3$ | $52.1 \pm 5.1$ | $52.0 \pm 5.1$ | $51.8 \pm 5.0$ |
| ExPerior-Param ($\beta = 0.1$) | $11.1 \pm 4.3$ | $33.1 \pm 7.3$ | $\mathbf{12.6 \pm 3.5}$ | $11.7 \pm 3.8$ | $10.9 \pm 4.2$ | $40.1 \pm 9.6$ | $12.3 \pm 4.7$ | $11.4 \pm 4.0$ | $10.7 \pm 4.2$ |
| ExPerior-MaxEnt ($\beta = 1$) | $45.7 \pm 3.4$ | $52.2 \pm 5.3$ | $51.6 \pm 5.1$ | $50.0 \pm 4.8$ | $47.3 \pm 3.8$ | $52.5 \pm 5.3$ | $51.0 \pm 4.8$ | $49.1 \pm 4.2$ | $48.0 \pm 3.6$ |
| ExPerior-Param ($\beta = 1$) | $9.1 \pm 3.0$ | $\mathbf{21.3 \pm 1.3}$ | $13.4 \pm 2.9$ | $\mathbf{10.1 \pm 3.0}$ | $9.4 \pm 3.1$ | $\mathbf{22.8 \pm 1.3}$ | $\mathbf{9.8 \pm 3.0}$ | $\mathbf{8.6 \pm 2.7}$ | $\mathbf{8.8 \pm 2.9}$ |
| ExPerior-MaxEnt ($\beta = 2.5$) | $\mathbf{37.0 \pm 1.9}$ | $52.1 \pm 5.3$ | $51.0 \pm 4.9$ | $47.1 \pm 4.5$ | $38.3 \pm 2.0$ | $\mathbf{52.1 \pm 5.1}$ | $48.9 \pm 4.1$ | $44.8 \pm 3.2$ | $40.5 \pm 2.1$ |
| ExPerior-Param ($\beta = 2.5$) | $\mathbf{8.5 \pm 2.8}$ | $24.3 \pm 1.2$ | $19.0 \pm 2.1$ | $12.8 \pm 2.9$ | $\mathbf{9.2 \pm 3.1}$ | $24.6 \pm 1.2$ | $15.9 \pm 3.0$ | $10.9 \pm 3.2$ | $\mathbf{8.8 \pm 2.9}$ |
| ExPerior-MaxEnt ($\beta = 10$) | $38.5 \pm 9.4$ | $\mathbf{52.0 \pm 5.2}$ | $\mathbf{47.6 \pm 4.4}$ | $\mathbf{39.7 \pm 2.9}$ | $\mathbf{29.7 \pm 3.6}$ | $52.5 \pm 5.3$ | $\mathbf{41.9 \pm 2.6}$ | $\mathbf{37.7 \pm 2.8}$ | $\mathbf{31.9 \pm 3.0}$ |
| ExPerior-Param ($\beta = 10$) | $11.2 \pm 4.8$ | $26.9 \pm 1.2$ | $25.0 \pm 1.5$ | $21.0 \pm 2.1$ | $11.8 \pm 3.3$ | $26.8 \pm 1.1$ | $23.2 \pm 1.8$ | $20.1 \pm 2.5$ | $16.1 \pm 3.0$ |
| Oracle-TS | $8.5 \pm 2.7$ | $8.5 \pm 2.7$ | $8.5 \pm 2.7$ | $8.5 \pm 2.7$ | $8.5 \pm 2.7$ | $8.5 \pm 2.7$ | $8.5 \pm 2.7$ | $8.5 \pm 2.7$ | $8.5 \pm 2.7$ |
| Oracle-TS (SGLD) | $24.2 \pm 3.9$ | $24.2 \pm 3.9$ | $24.2 \pm 3.9$ | $24.2 \pm 3.9$ | $24.2 \pm 3.9$ | $24.2 \pm 3.9$ | $24.2 \pm 3.9$ | $24.2 \pm 3.9$ | $24.2 \pm 3.9$ |

strations, where unobserved factors minimally affect the optimal action, should exhibit near-zero regret since there is no diversity in the unobserved contexts, and the experts are near-optimal. Conversely, in scenarios where unobserved factors significantly influence the optimal actions, we anticipate the regret to align with standard online regret bounds, similar to the outcomes of Thompson sampling with a non-informative prior. We conduct trials with ExPerior and `Oracle-TS` across various numbers of arms over $T = 1{,}500$ episodes, calculating the mean and standard error of Bayesian regret across distinct prior distributions. As depicted in Figure 3 (a), both ExPerior and `Oracle-TS` yield sub-linear regret relative to $K$ and $T$, comparable to the established regret bound of $\mathcal{O}(\sqrt{KT})$ for Thompson sampling. However, the middle panel indicates that the regret of ExPerior is proportional to the entropy of the optimal action, having an almost *linear* relationship. This observation seems to be in contrast with the standard Bayesian regret bounds for Thompson sampling under correct prior that have shown a sublinear relationship of $\mathcal{O}\left(\sqrt{\mathrm{Ent}(\pi_c)}\right)$, where $\mathrm{Ent}(\pi_c)$ denotes the entropy of the optimal action under $\mu^\star$ [50]. We analyze this observation in section 5.1.

**Ablations.** We also run additional experiments to assess the robustness of ExPerior to misspecified experts. We create expert data from different experts with various competence levels, such as optimal, noisily rational, and random-optimal experts, where the latter chooses an action optimally with a fixed probability and randomly otherwise. Table 1 shows ExPerior's robustness to different expert models. Setting $\beta = 10$ for training `ExPerior-MaxEnt` and $\beta = 1$ for `ExPerior-Param` achieves consistent out-performance among different expert types. Moreover, We evaluate the advantage of learning nonparametric max-entropy prior over misspecified parametric priors in Table 2. Even though `ExPerior-Param` with a Beta prior outperforms `ExPerior-MaxEnt`, `ExPerior-MaxEnt` is superior to `ExPerior-Param` if the prior mismatches the correct form (e.g., Gaussian or Gamma).

## 5.1 An Alternative Frequentist Approach for $K$-armed Bandits

To analyze the effect of expert data on the Bayesian regret, we devise an alternative *frequentist* approach, based on the successive elimination algorithm [51], which follows a similar intuition to Experts-as-Priors. In particular, we prove a bound on its Bayesian regret and show that the derived bound is proportional to a term that closely resembles the entropy of the optimal action, showing that the observation in the middle panel of Figure 3 (a) is consistent within different approaches.

Table 2: Superiority of `ExPerior-MaxEnt` compared to `ExPerior-Param` with misspecified parametric prior.

| | ExPerior-Param | ExPerior-MaxEnt | Gamma Prior | Beta-SGLD Prior | Normal Prior | Oracle-TS | Oracle-TS (SGLD) |
|---|---|---|---|---|---|---|---|
| **Low Entropy** | $0.7 \pm 0.3$ | $11.6 \pm 1.3$ | $39.3 \pm 2.2$ | $60.2 \pm 6.3$ | $546.5 \pm 153.4$ | $0.9 \pm 0.4$ | $11.0 \pm 1.6$ |
| **Mid-Entropy** | $6.8 \pm 0.8$ | $25.7 \pm 1.2$ | $36.8 \pm 0.9$ | $40.4 \pm 2.0$ | $492.5 \pm 185.6$ | $7.3 \pm 0.8$ | $21.2 \pm 1.0$ |
| **High-Entropy** | $24.5 \pm 2.8$ | $41.3 \pm 2.2$ | $51.8 \pm 3.6$ | $45.6 \pm 2.0$ | $461.8 \pm 104.8$ | $21.5 \pm 2.2$ | $39.9 \pm 3.2$ |

---

**Algorithm 2** Successive Elimination with Expert Sampling

---

1: **Input:** Episodes $T$, Arms $\mathcal{A}$, expert policy $\widehat{P}_E$, step size $p_{\min}$, unknown $c \sim \mu^\star$, and $\delta \in (0, 1)$.
2: **for** $t = 1 \ldots T$ **do**
3:     try an active arm $a$ with a relative frequency of $\lceil \frac{\widehat{P}_E(a)}{p_{\min}} \rceil$.          // all arms are active at $t = 0$.
    // $n_t(a)$ is the number of times arm $a$ is pulled by episode $t$ and $\overline{V_c^t}(a)$ is its empirical mean reward.
4:     increment $n_t(a)$ and update $\overline{V_c^t}(a)$.
5:     construct $\text{UCB}_a^t = \overline{V_c^t}(a) + \sqrt{\log\left(4T^4K/\delta\right)/2n_t(a)}$ and $\text{LCB}_a^t = \overline{V_c^t}(a) - \sqrt{\log\left(4T^4K/\delta\right)/2n_t(a)}$.
6:     de-activate all arms $a$ s.t. $\exists a'$ with $\text{UCB}_a \le \text{LCB}_{a'}$, and normalize $\widehat{P}_E$.
7: **end for**

---

The idea of successive elimination is to identify suboptimal arms and deactivate them over time. In particular, it runs a uniform sampling policy among active arms and builds confidence intervals for each. It then deactivates all the arms with an upper confidence bound smaller than at least one arm's lower confidence bound. We modify this algorithm using the policy derived from expert demonstrations instead of a uniform sampling policy. Recall that in $K$-armed bandits, each expert trajectory $\tau_E$ represents the pulled arm by the expert. Hence, the empirical distribution of expert demonstrations can be seen as a sampling policy over different arms. To simplify the analysis, we employ a deterministic sampling approach by pulling each arm a fixed number of times based on its probability. To do so, we discretize the expert policy with a step size $p_{\min}$, which leads to a relative frequency of $\lceil \widehat{P}_E(a)/p_{\min} \rceil$ for an arm $a$. In particular, we can choose $p_{\min} = \min_{a;\, \widehat{P}_E(a) \ne 0} \widehat{P}_E(a)$. We provide the concrete algorithm in Algorithm 2 and prove the following Bayesian regret bound:

**Theorem 2.** *Consider a stochastic $K$-armed bandit and let $p$ be the empirical expert policy. Assume that (i) the mean reward function is bounded in $[0, 1]$ for all arms, (ii) $T \ge \frac{1}{\min_{a;p(a)\ne 0} p(a)}$, (iii) the expert is optimal, i.e., $\forall a \in \mathcal{A}: \ p(a) = P_E(a;\mu^\star)$ and $\beta \to \infty$, and (iv) the learner follows Algorithm 2. Then, with probability at least $1 - \delta$,*

$$Reg \lesssim \sqrt{T\log\left(TK/\delta\right)} \sum_{a,a' \in \mathcal{A}, a \ne a'} \sqrt{\frac{p(a)}{p(a)+p(a')}\left(1 - \frac{p(a)}{p(a)+p(a')}\right)} \left[\sqrt{p(a)} + \sqrt{p(a')}\right]. \quad (4)$$

See Appendix A.4 for the proof. Two terms in (4) depend on expert data: (1) The relative standard deviation between any two pairs of arms and (2) a scaling factor that depends on the magnitude of probability that the arms are optimal. For homogeneous demonstrations, where the expert data only includes one unique pulled arm, the standard deviation (Term 1) is zero, resulting in zero regret. However, in extreme heterogeneity, where the empirical expert distribution is uniform over the arms, we have $Reg \lesssim K\sqrt{KT\log T}$. [2] Finally, to assess the relationship between the regret bound and the entropy of the expert data, we fix $K = 2$, $T = 100$, and plot the bound from (4) as a function of the entropy of the optimal action for various prior distributions. Figure 3 (b) demonstrates a linear relationship, similar to the regret incurred by ExPerior in Figure 3 (a). This observation opens up new directions to further analyze the regret for ExPerior and similar approaches in MDPs.

## 6 Learning in Markov Decision Processes (MDPs)

For MDPs, we need to parameterize both the mean reward and transition functions. However, we assume the transition functions are invariant to the context variables to simplify our methodology and avoid extra modelling assumptions. Under this assumption, it is sufficient to parameterize the *optimal* Q-functions, e.g., using a deep Q-network (DQN) and treat those parameters as a proxy for the unobserved context variables, i.e., $\mathcal{C}_{\text{MDP}} := \{(s, a) \mapsto Q(s, a;\boldsymbol{\theta}) ; \boldsymbol{\theta} \in \boldsymbol{\Theta}\}$, where $\boldsymbol{\Theta}$ is the set of parameters for a DQN. We can then derive a closed-form log-pdf of the posterior distribution under the maximum entropy prior. See Appendix A.5 for details. The derived posterior log-pdf can

---

[2]Although this bound is worse than the standard successive elimination by a factor of $K$, our empirical results show that ExPerior is still on par with standard regrets in the non-informative cases.

Table 3: The average reward per episode in Frozen Lake (PODMP) after 90,000 training steps.

| | Fixed # Hazard $= 9$ | | | | Fixed $\beta = 1$ | | | |
|---|---|---|---|---|---|---|---|---|
| | $\beta = 0.1$ | $\beta = 1$ | $\beta = 2.5$ | $\beta = 10$ | # Hazard $= 2$ | # Hazard $= 5$ | # Hazard $= 7$ | # Hazard $= 9$ |
| **(POMDP)** | | | | | | | | |
| ExPerior-MaxEnt | -22.58 ± 1.17 | **6.00 ± 0.00** | 3.58 ± 0.89 | 1.62 ± 1.85 | 11.47 ± 0.52 | **5.71 ± 0.67** | **6.00 ± 0.00** | **6.00 ± 0.00** |
| ExPerior-Param | -23.32 ± 0.69 | -4.31 ± 1.80 | 5.27 ± 0.51 | **6.00 ± 0.00** | **12.00 ± 0.37** | 2.11 ± 1.41 | 5.42 ± 0.40 | -4.31 ± 1.80 |
| Naïve Boot-DQN | -23.32 ± 0.69 | -23.32 ± 0.69 | -23.32 ± 0.69 | -23.32 ± 0.69 | -14.36 ± 5.88 | -20.57 ± 2.91 | -20.39 ± 1.75 | -23.32 ± 0.69 |
| ExPLORe | **5.99 ± 0.00** | **6.00 ± 0.00** | **6.00 ± 0.00** | **6.00 ± 0.00** | -30.68 ± 12.40 | -10.64 ± 16.64 | -13.00 ± 19.00 | **6.00 ± 0.00** |
| Optimal | 6.00 ± 0.00 | 6.00 ± 0.00 | 6.00 ± 0.00 | 6.00 ± 0.00 | 12.00 ± 0.37 | 6.53 ± 0.31 | 6.00 ± 0.00 | 6.00 ± 0.00 |
| **(MDP)** | | | | | | | | |
| ExPerior-MaxEnt | -23.36 ± 1.26 | 12.26 ± 0.29 | 12.68 ± 0.03 | **12.71 ± 0.03** | **13.02 ± 0.18** | **12.78 ± 0.11** | **12.78 ± 0.06** | 12.26 ± 0.29 |
| ExPerior-Param | -25.53 ± 2.35 | **12.64 ± 0.08** | **12.70 ± 0.03** | 12.68 ± 0.03 | 13.00 ± 0.18 | **12.78 ± 0.12** | 12.73 ± 0.07 | **12.64 ± 0.08** |
| Naïve Boot-DQN | -23.32 ± 0.69 | -23.32 ± 0.69 | -23.32 ± 0.69 | -23.32 ± 0.69 | -14.39 ± 5.22 | -20.99 ± 2.86 | -20.39 ± 1.75 | -23.32 ± 0.69 |
| ExPLORe | **11.74 ± 0.41** | 11.75 ± 0.63 | 11.96 ± 0.28 | 12.3 ± 0.22 | -113.84 ± 17.50 | -54.89 ± 13.75 | -10.00 ± 7.60 | 11.75 ± 0.63 |
| Optimal | 12.71 ± 0.03 | 12.71 ± 0.03 | 12.71 ± 0.03 | 12.71 ± 0.03 | 13.02 ± 0.18 | 12.78 ± 0.11 | 12.76 ± 0.06 | 12.64 ± 0.03 |

then be used as the loss function for DQN Langevin Monte Carlo [52, 53] as the counterpart for Thompson sampling with SGLD. However, running Langevin dynamics can lead to highly unstable policies due to the complexity of the optimization landscape in DQNs. Instead, we use a heuristic that combines the learned prior distribution with bootstrapped DQNs [54].

The original method of Bootstrapped DQNs utilizes an ensemble of $L$ randomly initialized Q-networks. It samples a Q-network uniformly at each episode and uses it to collect data. Then, each Q-network is trained using the temporal difference loss on parts of or possibly the entire collected data. This method and its subsequent iterations [55, 56, 57] achieve deep exploration by ensuring diversity among the learned Q-networks. To incorporate Bootstrapped DQN into the ExPerior framework and utilize the expert data, we can formulate the ensemble as a discrete prior distribution over the Q-networks. Let $\boldsymbol{\theta}_{\text{ens}} = \left(\boldsymbol{\theta}_{\text{ens}}^1, \ldots, \boldsymbol{\theta}_{\text{ens}}^L\right)$ be the parameter vector for an ensemble of Q-functions. We can define the ensemble prior, parameterized by $\boldsymbol{\theta}_{\text{ens}}$, as $\mu_{\boldsymbol{\theta}_{\text{ens}}}(\boldsymbol{\theta}) := \frac{1}{L} \sum_{i=1}^{L} \mathbb{I}\left(\boldsymbol{\theta}_{\text{ens}}^i = \boldsymbol{\theta}\right)$ for any $\boldsymbol{\theta} \in \boldsymbol{\Theta}$. Based on this prior model, we can learn the parametric expert prior using maximum marginal likelihood estimation, as formulated below.

**Proposition 3** (Ensemble Marginal Likelihood). *Consider a contextual MDP $\mathcal{M} = (\mathcal{S}, \mathcal{A}, \mathcal{T}, R, H, \rho, \mu^\star)$. Assume the transition function $\mathcal{T}$ does not depend on the context variables and Assumption 1 holds. Then, the negative marginal log-likelihood of expert data $\mathcal{D}_E$ under the ensemble prior $\mu_{\boldsymbol{\theta}_{\text{ens}}}$ is upper bounded by*

$$-\log \mathrm{P}_E\left(\mathcal{D}_E \,;\, \mu_{\boldsymbol{\theta}_{\text{ens}}}\right) \leq \frac{1}{L} \sum_{i=1}^{L} \sum_{\tau \in \mathcal{D}_E} \sum_{(s,a) \in \tau} \log \left( \sum_{a' \in \mathcal{A}} \exp \left\{ \beta \cdot Q\left(s, a' \,;\, \boldsymbol{\theta}_{\text{ens}}^i\right) \right\} \right) - \beta \cdot Q\left(s, a \,;\, \boldsymbol{\theta}_{\text{ens}}^i\right),$$

*where $\beta$ is the competence level of the expert in Assumption 1.*

Proposition 3 is proved in Appendix A.6. We can then initialize the Q-networks in the Bootstrapped DQN method using ensemble parameters that minimize the above upper bound. We will refer to this method as `ExPerior-Param`. As an alternative approach, instead of minimizing the above upper bound, we can match the discrete prior distribution $\mu_{\boldsymbol{\theta}_{\text{ens}}}$ to the max-entropy prior by initializing the Q-functions in the ensemble with parameters sampled from the max-entropy expert prior. In particular, we can apply SGLD on the log-pdf of the max-entropy prior derived in Appendix A.5. We will refer to this approach as `ExPerior-MaxEnt`.

**Experimental Setup.** One challenge in RL is the reward *sparsity*, where the learner needs to explore the environment deeply to observe rewards. Utilizing expert demonstrations can significantly improve the efficiency of exploration. Here, we focus on "Deep Sea," a sparse-reward tabular RL environment proposed by Osband et al. [56] to assess deep exploration for different RL methods. The environment is an $M \times M$ grid, where the agent starts at the top-left corner of the map, and at each time step, it chooses an action from $\mathcal{A} = \{\texttt{left}, \texttt{right}\}$ to move to the left or right column, while going down by one row. In the original version of Deep Sea, the goal is always on the bottom-right corner of the map. We introduce unobserved contexts by defining a distribution over the goal columns while keeping the goal row the same. We consider four types of goal distributions where the goal is situated at (1) the bottom-right corner of the grid, (2) uniformly at the bottom of any of the right-most $\frac{M}{4}$ columns, (3) uniformly at the bottom of any of the right-most $\frac{M}{2}$ columns, and (4) uniformly at the bottom of any of the $M$ columns. We set $M = 30$ and generate $N = 1,000$

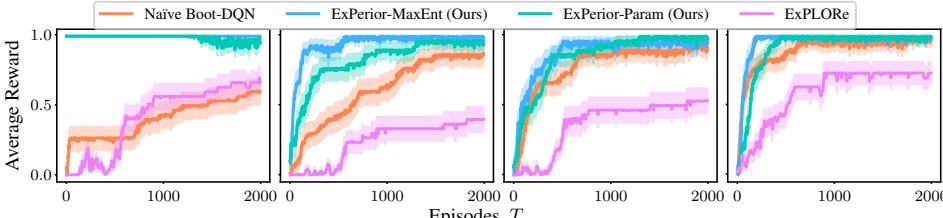

Figure 4: The average reward per episode over 2,000 episodes in "Deep Sea." The goal is located at the right column, uniformly at the right-most quarter of the columns, uniformly at the right-most half, and uniformly at random over all the columns, respectively. ExPerior outperforms the baselines in all instances.

samples from the optimal policies as offline expert demonstrations. To further evaluate ExPerior and showcase its applicability to partially-observed MDP, we also consider the "Frozen Lake" environment, which requires the learner to navigate to a goal while avoiding hazards [17]. The learner cannot observe the hazard location, while the expert has access to the whole map. Taking action, reaching the goal, and hitting the hazard incur rewards of -2, 20, and -100, respectively. The frozen lake map is $5 \times 5$, where the hazard (weak ice) is randomly located in the interior squares. We consider different settings with 2, 5, 7, and 9 potential locations for the hazard. At the start of each episode, the hazard will be chosen randomly within the potential locations. We generate $N = 1,000$ samples from noisily rational experts with different competence levels for this environment.

**Baselines.** We compare ExPerior to the following: (1) `ExPLORe`, proposed by Li et al. [38] to accelerate off-policy reinforcement learning using unlabeled prior data. In this method, the offline demonstrations are assigned optimistic reward labels generated using the online data with regular updates. This information is then combined with the buffer data to perform off-policy learning. (2) `Naïve Boot-DQN`, which is the original Bootstrapped DQN with randomly initialized Q-networks [54].

**Deep Sea Results.** Figure 4 demonstrates the average reward per episode achieved by the baselines for $T = 2,000$ episodes. For each goal distribution, we run the baselines with 30 different seeds and take the average to estimate the expected reward. ExPerior outperforms the baselines in all instances. However, the gap between ExPerior and the fully online `Naïve Boot-DQN`, which measures the effect of using the expert data, decreases as we go from the low-entropy setting (upper left) to the high-entropy distribution over the contexts (bottom right). This is consistent with the empirical and theoretical results discussed in section 5 and confirms our expectation that the expert demonstrations may not be helpful under strong unobserved confounding (strong heterogeneity). The `ExPLORe` baseline substantially underperforms, even compared to the fully online `Naïve Boot-DQN` (except for the first distribution with zero-entropy). We suspect this is because `ExPLORe` uses actor-critic methods as its backbone model, which are shown to struggle with deep exploration [58].

**Frozen Lake Results.** We run all the baselines for 90,000 steps with 30 different seeds. Table 3 shows the average reward after 500 evaluation steps at the end of the training. ExPerior outperforms the baselines in almost all instances except for the case of $\beta = 0.1$, which corresponds to a nearly random expert. On the other hand, `ExPLORe` achieves near-optimal results for $\beta = 0.1$. We hypothesize that `ExPLORe`'s performance is mainly due to the superiority of their base actor-critic model since it can achieve near-optimal performance even when the expert trajectories are low-quality.

## 7  Conclusion

We introduce the Experts-as-Priors (ExPerior) framework, a novel empirical Bayes approach to address the problem of sequential decision-making using expert demonstrations with unobserved heterogeneity. We ground our methodology in the maximum entropy principle to infer a prior distribution from expert data that guides the learning process in different setttings, including bandits, Markov decision processes (MDPs), and partially-observed MDPs. Our experimental evaluations demonstrate that we can effectively leverage the expert demonstrations to enhance learning efficiency under unobserved confounding. In multi-armed bandits, we illustrated through empirical analysis that the Bayesian regret incurred by our method is proportional to the entropy of the optimal action, highlighting its capacity to adapt based on the informativeness of the expert data. Our work offers a practical framework readily applied to a broad spectrum of decision-making tasks. One limitation of our work is the limited set of experiments, especially the lack of experiments with human-in-the-loop. Future directions include extending to more complex environments and further investigating the theoretical properties of our RL algorithm.

## Acknowledgements

We would like to thank Benjamin Van Roy, Emma Brunskill, Florian Shkurti, Ramesh Johari, and Sanath Kumar Krishnamurthy for their valuable discussions and insights. We also acknowledge the use of GPT-4 in Figure 1 and for assistance in editing various sections of this manuscript. KC is supported by the William R. and Sara Hart Kimball Stanford Graduate Fellowship. VS is supported by NSF Award IIS-2337916. RGK is supported by a Canada CIFAR AI Chair. VBM and VN were supported by an NSERC Discovery Award RGPIN-2022-04546 and an NFRF Special Call NFRFR-2022-00526. Resources used in preparing this research were provided, in part, by the Province of Ontario, the Government of Canada through CIFAR, and companies sponsoring the Vector Institute.

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

# A   Proofs

## A.1   Notation

We assume $\mathcal{C}$ is a measurable set with an appropriate $\sigma$-algebra and there exists a probability measure $\mu_0$ on $\mathcal{C}$. We denote $L^p(\mathcal{C}, \mu_0)$ as the space of all measurable functions $f : \mathcal{C} \to \mathbb{R}$ such that $\|f\|_p = \left( \int_{\mathcal{C}} |f|^p \, \mathrm{d}\mu_0 \right)^{1/p} < \infty$. Moreover, we define $L^\infty(\mathcal{C}, \mu_0)$ as the space of all essentially bounded measurable functions from $\mathcal{C}$ to $\mathbb{R}$. Unless stated otherwise, we assume the probability measures are absolutely continuous w.r.t. $\mu_0$, and their density functions are in $L^1(\mathcal{C}, \mu_0)$. We may abuse the notation and use the same symbol for a probability measure and its Radon–Nikodym derivative w.r.t. $\mu_0$. Finally, we use $\mathbb{E}\left[\cdot\right]$ to denote expectation under the probability measure $\mu_0$.

## A.2   Useful Lemmas

Here, we state and prove a set of results that will be useful for the rest of this section. The first one is Fenchel's duality theorem:

**Lemma 4** (Fenchel's Duality [59]). *Let $X$ and $Y$ be Banach spaces, let $f : X \to \mathbb{R} \cup \{+\infty\}$ and $g : Y \to \mathbb{R} \cup \{+\infty\}$ be convex functions and let $A : X \to Y$ be a bounded linear map. Define the primal and dual values $p, d \in [-\infty, +\infty]$ by the Fenchel problems*

$$p = \inf_{x \in X} \; f(x) + g(Ax)$$
$$d = \sup_{y^* \in Y^*} \; -f^*(A^* y^*) - g^*(-y^*),$$

*where $f^*$ and $g^*$ are the Fenchel conjugates of $f$ and $g$ defined as $f^*(x^*) = \sup_{x \in X} \langle x^*, x \rangle - f(x)$ (similarly for $g$), $X^*$ is the dual space of $X$ and $\langle \cdot, \cdot \rangle$ is its duality pairing, and $A^* : Y^\star \to X^\star$ is the adjoint operator of $A$, i.e., $\langle A^* y^*, x \rangle = \langle y^*, A x \rangle$. Suppose $A \, dom(f) \cap cont(g) \neq \emptyset$, where $dom(f) := \{x \in X \, ; \, f(x) < \infty\}$ and $cont(g)$ are the continuous points of $g$. Then, strong duality holds, i.e., $p = d$.*

*Proof.* See the proof of Theorem 4.4.3 in Borwein and Zhu [59]. $\qquad\square$

We can use Fenchel's duality to solve generalized maximum entropy problems. In particular, we prove a generalization of Theorem 2 in [48] for density functions in $L^1(\mathcal{C}, \mu_0)$:

**Lemma 5.** *For any function $\mu \in L^1(\mathcal{C}, \mu_0)$, define the extended KL divergence as*

$$\psi(\mu) := \begin{cases} \mathrm{D}_{\mathrm{KL}}\left(\mu \,\|\, \mu_0\right) & \textit{If } \|\mu\|_1 = 1, \\ +\infty & \textit{o.w.} \end{cases}$$

*Moreover, assume a set of bounded feature functions $m_1, m_2, \ldots, m_N : \mathcal{C} \to \mathbb{R}$ is given and denote $\mathbf{m}$ as the vector of all $N$ features. Consider the linear function $A_{\mathbf{m}} : L^1(\mathcal{C}, \mu_0) \to \mathbb{R}^N$ defined as*

$$\forall \mu \in L^1(\mathcal{C}, \mu_0): \; A_{\mathbf{m}}(\mu) := \left( \mathbb{E}\left[m_1 \cdot \mu\right], \mathbb{E}\left[m_2 \cdot \mu\right], \ldots, \mathbb{E}\left[m_N \cdot \mu\right] \right).$$

*We define the generalized maximum entropy problem as the following:*

$$\inf_{\mu \in L^1(\mathcal{C}, \mu_0)} \psi(\mu) + \zeta\left(A_{\mathbf{m}}(\mu)\right), \tag{5}$$

*for an arbitrary closed proper convex function $\zeta : \mathbb{R}^N \to \mathbb{R}$. Then the following holds:*

1. *The dual optimization of (5) is given by*

$$\sup_{\boldsymbol{\alpha} \in \mathbb{R}^N} \; -\log \mathbb{E}\left[\exp\left\{\mathbf{m}^\top \boldsymbol{\alpha}\right\}\right] - \zeta^*\left(-\boldsymbol{\alpha}\right), \tag{6}$$

   *where $\zeta^\star$ is the convex conjugate function of $\zeta$.*

2. *Denote $\boldsymbol{\alpha}^1, \boldsymbol{\alpha}^2, \ldots$ as a sequence in $\mathbb{R}^N$ converging to supremum (6), and define the following Gibbs density functions*

$$\mu_{Gibbs}^{\boldsymbol{\alpha}}(c) := \frac{\exp\left\{\mathbf{m}(c)^\top \boldsymbol{\alpha}\right\}}{\mathbb{E}\left[\exp\left\{\mathbf{m}^\top \boldsymbol{\alpha}\right\}\right]}.$$

   *Then,*

$$\inf_{\mu \in L^1(\mathcal{C}, \mu_0)} \psi(\mu) + \zeta\left(A_{\mathbf{m}}(\mu)\right) = \lim_{n \to \infty} \psi(\mu_{Gibbs}^{\boldsymbol{\alpha}^n}) + \zeta\left(A_{\mathbf{m}}(\mu_{Gibbs}^{\boldsymbol{\alpha}^n})\right).$$

*Proof.* **Part 1:** We first derive the convex conjugate of $\psi$. Note that $\left(L^1(\mathcal{C}, \mu_0)\right)^\star = L^\infty(\mathcal{C}, \mu_0)$ with the pairing

$$\forall h \in L^\infty(\mathcal{C}, \mu_0),\ \mu \in L^1(\mathcal{C}, \mu_0): \ \langle h, \mu \rangle := \int_{\mathcal{C}} h(c) \cdot \mu(c)\, \mathrm{d}\mu_0.$$

Hence, by Donsker and Varadhan's variational formula

$$\forall h \in L^\infty(\mathcal{C}, \mu_0): \ \psi^\star(h) = \sup_{\mu \in L^1(\mathcal{C}, \mu_0)} \langle h, \mu \rangle - \psi(\mu) = \log \mathbb{E}\left[\exp\left\{h\right\}\right]. \tag{7}$$

Moreover, the adjoint operator of $A_{\mathbf{m}}$ is given by $A_{\mathbf{m}}^\star : \mathbb{R}^N \to (\mathcal{C} \to \mathbb{R})$:

$$\forall \boldsymbol{\alpha} \in \mathbb{R}^N,\ c \in \mathcal{C}: \ A_{\mathbf{m}}^\star(\boldsymbol{\alpha})(c) = \mathbf{m}(c)^\top \boldsymbol{\alpha}. \tag{8}$$

Using (7) and (8) and Lemma 4 concludes the proof.

**Part 2:** Denote the primal and dual objective functions by

$$P(\mu) := \psi(\mu) + \zeta\left(A_{\mathbf{m}}(\mu)\right),$$
$$D(\boldsymbol{\alpha}) := -\log \mathbb{E}\left[\exp\left\{\mathbf{m}^\top \boldsymbol{\alpha}\right\}\right] - \zeta^*(-\boldsymbol{\alpha}),$$

and their optimal values as $P^*$ and $D^*$. For any $\nu \in L^1(\mathcal{C}, \mu_0)$, note that

$$\begin{aligned}
\mathrm{D}_{\mathrm{KL}}\left(\nu \,\|\, \mu_0\right) - \mathrm{D}_{\mathrm{KL}}\left(\nu \,\|\, \mu_{\mathrm{Gibbs}}^{\boldsymbol{\alpha}}\right) &= \int_{\mathcal{C}} \nu \log \nu\, \mathrm{d}\mu_0 - \left(\int_{\mathcal{C}} \nu \log \nu\, \mathrm{d}\mu_0 - \int_{\mathcal{C}} \nu \log \mu_{\mathrm{Gibbs}}^{\boldsymbol{\alpha}}\, \mathrm{d}\mu_0\right) \\
&= \int_{\mathcal{C}} \left(\mathbf{m}(c)^\top \boldsymbol{\alpha}\right) \nu(c)\, \mathrm{d}\mu_0 - \log \mathbb{E}\left[\exp\left\{\mathbf{m}^\top \boldsymbol{\alpha}\right\}\right] \\
&= A_{\mathbf{m}}(\nu)^\top \boldsymbol{\alpha} - \log \mathbb{E}\left[\exp\left\{\mathbf{m}^\top \boldsymbol{\alpha}\right\}\right]. 
\end{aligned} \tag{9}$$

Using (9), we can re-write the dual objective function as:

$$\forall \boldsymbol{\alpha} \in \mathbb{R}^N, \nu \in L^1(\mathcal{C}, \mu_0): \quad D(\boldsymbol{\alpha}) = -\mathrm{D}_{\mathrm{KL}}\left(\nu \,\|\, \mu_{\mathrm{Gibbs}}^{\boldsymbol{\alpha}}\right) + \mathrm{D}_{\mathrm{KL}}\left(\nu \,\|\, \mu_0\right) - A_{\mathbf{m}}(\nu)^\top \boldsymbol{\alpha} - \zeta^\star(-\boldsymbol{\alpha}). \tag{10}$$

Moreover, note that

$$\begin{aligned}
-A_{\mathbf{m}}(\nu)^\top \boldsymbol{\alpha} - \zeta^\star(-\boldsymbol{\alpha}) &= -A_{\mathbf{m}}(\nu)^\top \boldsymbol{\alpha} - \left(\sup_x \langle x, -\boldsymbol{\alpha}\rangle - \zeta(x)\right) \\
&\leq -A_{\mathbf{m}}(\nu)^\top \boldsymbol{\alpha} - \left(\langle A_{\mathbf{m}}(\nu), -\alpha\rangle - \zeta(A_{\mathbf{m}}(\nu))\right) \\
&= \zeta(A_{\mathbf{m}}(\nu)).
\end{aligned} \tag{11}$$

Combining (10) and (11), we get

$$\begin{aligned}
\forall \boldsymbol{\alpha} \in \mathbb{R}^N, \nu \in L^1(\mathcal{C}, \mu_0): \quad D(\boldsymbol{\alpha}) &\leq -\mathrm{D}_{\mathrm{KL}}\left(\nu \,\|\, \mu_{\mathrm{Gibbs}}^{\boldsymbol{\alpha}}\right) + \mathrm{D}_{\mathrm{KL}}\left(\nu \,\|\, \mu_0\right) + \zeta(A_{\mathbf{m}}(\nu)) \\
&= -\mathrm{D}_{\mathrm{KL}}\left(\nu \,\|\, \mu_{\mathrm{Gibbs}}^{\boldsymbol{\alpha}}\right) + P(\nu).
\end{aligned} \tag{12}$$

Now, fix an arbitrary $\epsilon > 0$, and consider a sequence of $\mu^1, \mu^2, \ldots \in L^1(\mathcal{C}, \mu_0)$ such that for all $j \in \mathbb{N}$:

$$P(\mu^j) - P^* < \frac{\epsilon}{2^j}. \tag{13}$$

We can re-write (13) using the fact $P^* = D^* = \lim_{n \to \infty} D(\boldsymbol{\alpha}^n)$:

$$\forall j \in \mathbb{N}: \quad \lim_{n \to \infty} P(\mu^j) - D(\boldsymbol{\alpha}^n) < \frac{\epsilon}{2^j} \tag{14}$$

In particular, by setting $\nu = \mu^j$ in (12) and combining the result with (14), we get

$$\forall j \in \mathbb{N}: \quad \lim_{n \to \infty} \mathrm{D}_{\mathrm{KL}}\left(\mu^j \,\|\, \mu_{\mathrm{Gibbs}}^{\boldsymbol{\alpha}^n}\right) < \frac{\epsilon}{2^j}.$$

Hence, $\lim_{j \in \infty} \lim_{n \to \infty} \mathrm{D}_{\mathrm{KL}}\left(\mu^j \,\|\, \mu_{\mathrm{Gibbs}}^{\boldsymbol{\alpha}^n}\right) = 0$. From properties of the KL divergence, it follows that $\lim_{j \to \infty} P(\mu^j) = \lim_{n \to \infty} P(\mu_{\mathrm{Gibbs}}^{\boldsymbol{\alpha}^n})$, concluding the proof. $\qquad\square$

### A.3  Max-Entropy Prior

**Proposition 1.** *Let $N = |\mathcal{D}_E|$ be the number of demonstrations in $\mathcal{D}_E$. For each $c \in \mathcal{C}$ and demonstration $\tau_E = (s_1, a_1, s_2, a_2, \ldots, s_H, a_H, s_{H+1}) \in \mathcal{D}_E$, define $m_{\tau_E}(c)$ as the (partial) likelihood of $\tau_E$ under $c$:*

$$m_{\tau_E}(c) = \prod_{h=1}^{H} p_E\left(a_h \mid s_h\,;\,c\right) \mathcal{T}\left(s_{h+1} \mid s_h, a_h, c\right). \tag{15}$$

*Denote $\mathbf{m}(c) \in \mathbb{R}^N$ as the vector with elements $m_{\tau_E}(c)$ for $\tau_E \in \mathcal{D}_E$. Moreover, let $\lambda^\star \in \mathbb{R}^{\geq 0}$ be the optimal solution to the Lagrange dual problem of (2). Then, the solution to optimization (2) is as follows:*

$$\mu_{ME}(c) = \lim_{n \to \infty} \frac{\exp\left\{\mathbf{m}(c)^\top \boldsymbol{\alpha}_n\right\}}{\mathbb{E}_{c \sim \mu_0}\left[\exp\left\{\mathbf{m}(c)^\top \boldsymbol{\alpha}_n\right\}\right]},$$

*where $\{\boldsymbol{\alpha}_n\}_{n=1}^{\infty}$ is a sequence converging to the following supremum:*

$$\sup_{\boldsymbol{\alpha} \in \mathbb{R}^N} -\log \mathbb{E}_{c \sim \mu_0}\left[\exp\left\{\mathbf{m}(c)^\top \boldsymbol{\alpha}\right\}\right] + \frac{\lambda^\star}{N} \sum_{i=1}^{N} \log\left(\frac{N \cdot \alpha_i}{\lambda^\star}\right).$$

*Proof.* We first simplify the KL-divergence between the empirical distribution of the expert trajectories $\widehat{P}_E$ and the marginal likelihood $P_E\left(\,\cdot\,;\,\mu\right)$:

$$
\begin{aligned}
D_{KL}\left(\widehat{P}_E \,\big\|\, P_E\left(\,\cdot\,;\,\mu\right)\right) &= \sum_{\tau^{(i)} \in \mathcal{D}_E} \widehat{P}_E(\tau^{(i)}) \log \frac{\widehat{P}_E(\tau^{(i)})}{P_E\left(\tau^{(i)}\,;\,\mu\right)} \\
&= -\log N - \frac{1}{N} \sum_{\tau^{(i)} \in \mathcal{D}_E} \log P_E\left(\tau^{(i)}\,;\,\mu\right) \qquad (\widehat{P}_E(\tau^{(i)}) = \tfrac{1}{N}) \\
&= -\log N - \frac{1}{N} \sum_{\tau^{(i)} \in \mathcal{D}_E} \log \mathbb{E}\left[m_{\tau^{(i)}} \cdot \mu\right] - \frac{1}{N} \sum_{s_1^{(i)} \in \mathcal{D}_E} \log \rho\left(s_1^{(i)}\right).
\end{aligned}
$$

$$\text{By (1) and (15)}$$

Using the above equality, we can re-write the definition of uncertainty set $\mathcal{P}(\epsilon)$ as

$$\mathcal{P}(\epsilon) = \left\{\mu\,;\, -\frac{1}{N} \sum_{\tau \in \mathcal{D}_E} \log \mathbb{E}\left[m_\tau \cdot \mu\right] - \epsilon - \log N - \frac{1}{N} \sum_{s_1 \in \mathcal{D}_E} \log \rho\left(s_1\right) \leq 0\right\}.$$

Therefore, we can re-write the optimization (2) as

$$\mu_{ME} = \operatorname*{arg\,min}_{\mu \in L^1(\mathcal{C}, \mu_0)} \psi(\mu) \quad \text{s.t.} \quad -\frac{1}{N} \sum_{\tau \in \mathcal{D}_E} \log \mathbb{E}\left[m_\tau \cdot \mu\right] - \epsilon - \log N - \frac{1}{N} \sum_{s_1 \in \mathcal{D}_E} \log \rho\left(s_1\right) \leq 0, \tag{16}$$

where the extended KL divergence $\psi(\mu)$ is defined as:

$$\psi(\mu) := \begin{cases} D_{KL}\left(\mu \,\|\, \mu_0\right) & \text{If } \|\mu\|_1 = 1, \\ +\infty & \text{o.w.} \end{cases}$$

Note that $\mathcal{P}(\epsilon)$ is a convex set. To see this, consider $\mu_1, \mu_2 \in \mathcal{P}(\epsilon)$. Then, for any $0 \leq \lambda \leq 1$, we have $\mu = (1 - \lambda)\mu_1 + \lambda \mu_2 \in \mathcal{P}(\epsilon)$ since $\mathbb{E}\left[m_\tau \cdot \mu\right]$ is linear in $\mu$ and $-\log$ is convex. Moreover, It is easy to see there exists a strictly feasible solution for (16) (e.g., consider the true distribution $\mu^\star$ over $\mathcal{C}$). Thus, strong duality holds, and we can form the Lagrangian function as

$$L(\mu, \lambda) := \psi(\mu) + \lambda\left(\frac{1}{N} \sum_{\tau \in \mathcal{D}_E} -\log \mathbb{E}\left[m_\tau \cdot \mu\right]\right) - \lambda\left(\epsilon + \log N + \frac{1}{N} \sum_{s_1 \in \mathcal{D}_E} \log \rho\left(s_1\right)\right).$$

Given that $\lambda^\star \in \mathbb{R}^{\geq 0}$ is the optimal solution to the Lagrange dual problem, the maximum entropy prior $\mu_{\mathrm{ME}}$ will be the solution to

$$\inf_{\mu \in L^1(\mathcal{C},\mu_0)} L(\mu, \lambda^\star) = \inf_{\mu \in L^1(\mathcal{C},\mu_0)} \psi(\mu) + \lambda^\star \left( \frac{1}{N} \sum_{\tau \in \mathcal{D}_{\mathrm{E}}} - \log \mathbb{E}\left[m_\tau \cdot \mu\right] \right) + \text{constant in } \mu. \quad (17)$$

Now, for each $\mathbf{x} \in \mathbb{R}^N$, define the convex function $\zeta(\mathbf{x}) := \frac{\lambda^\star}{N} \left( \sum_{i=1}^N - \log x_i \right)$. Moreover, for $\mu \in L^1(\mathcal{C},\mu_0)$, define $A_{\mathbf{m}}(\mu) := \left( \mathbb{E}\left[m_{\tau^{(1)}} \cdot \mu\right], \mathbb{E}\left[m_{\tau^{(2)}} \cdot \mu\right], \dots, \mathbb{E}\left[m_{\tau^{(N)}} \cdot \mu\right] \right)$. Then,

$$L(\mu, \lambda^\star) = \psi(\mu) + \zeta\left(A_{\mathbf{m}}(\mu)\right). \quad (18)$$

Combining (17) and (18), the maximum entropy prior $\mu_{\mathrm{ME}}$ is the solution to

$$\inf_{\mu \in L^1(\mathcal{C},\mu_0)} \psi(\mu) + \zeta\left(A_{\mathbf{m}}(\mu)\right).$$

Using Lemma 5 and noting that

$$\zeta^*(x^*) = \frac{\lambda^\star}{N} \left( \sum_{i=1}^N -1 - \log\left( -\frac{N}{\lambda^\star} \cdot x_i^* \right) \right)$$

concludes the proof. $\qquad\square$

### A.4 Regret Bound for $K$-armed Bandit

**Theorem 2.** *Consider a stochastic $K$-armed bandit and let $p$ be the empirical expert policy. Assume that (i) the mean reward function is bounded in $[0,1]$ for all arms, (ii) $T \geq \frac{1}{\min_{a;p(a)\neq 0} p(a)}$, (iii) the expert is optimal, i.e., $\forall a \in \mathcal{A} : p(a) = \mathrm{P}_E\left(a\,;\,\mu^\star\right)$ and $\beta \to \infty$, and (iv) the learner follows Algorithm 2. Then, with probability at least $1 - \delta$,*

$$Reg \lesssim \sqrt{T\log\left(TK/\delta\right)} \sum_{a,a' \in \mathcal{A}, a \neq a'} \sqrt{\frac{p(a)}{p(a) + p(a')} \left( 1 - \frac{p(a)}{p(a) + p(a')} \right)} \left[ \sqrt{p(a)} + \sqrt{p(a')} \right].$$

*Proof.* Fix $\delta \in (0,1)$ and $c \in \mathcal{C}$. Let $\mathcal{E}$ be the event that $\left| \overline{V}_c^t(a) - V_c(a) \right| \leq \sqrt{\frac{\log(4T^4K/\delta)}{2n_t(a)}}$ for all arms $a \in \mathcal{A}$, all $t \leq T$, and all $T \in \mathbb{N}$, where $n_t(a)$ is the number of times that arm $a$ was pulled by time $t$. Note that since $T \geq \frac{1}{p_{\min}}$, each arm will be pulled at least once and $n_t(a) \geq 1$.

We first show that $\mathbb{P}\left(\mathcal{E}\right) \geq 1 - \delta$. Fix $T$, arm $a$, and $t \leq T$. Suppose $n_t(a) = j$ for $1 \leq j \leq T$. By Hoeffding's inequality, we have

$$\mathbb{P}\left( \left| \overline{V}_c^t(a) - V_c(a) \right| \leq \sqrt{\frac{\log\left(4T^4K/\delta\right)}{2j}} \right) \geq 1 - \frac{\delta}{2T^4K}. \quad (19)$$

Now, using the union bound over all episodes and all actions, we get

$$\mathbb{P}\left( \exists a \in \mathcal{A}, T \in \mathbb{N}, t \leq T, j \leq t : \left| \overline{V}_c^t(a) - V_c(a) \right| > \sqrt{\frac{\log\left(2T^4K/\delta\right)}{2j}} \right)$$

$$\leq \sum_{T=1}^\infty \sum_{a \in \mathcal{A}} \sum_{t=1}^T \sum_{j=1}^t \mathbb{P}\left( \left| \overline{V}_c^t(a) - V_c(a) \right| > \sqrt{\frac{\log\left(2T^4K/\delta\right)}{2j}} \right)$$

$$\leq \sum_{T=1}^\infty \sum_{a \in \mathcal{A}} \sum_{t=1}^T t \cdot \frac{\delta}{2T^4K} \qquad\qquad \text{By (19)}$$

$$\leq \sum_{T=1}^\infty \frac{\delta}{2T^4K} \times T^2 \times K = \sum_{T=1}^\infty \frac{\delta}{2T^2} \leq \delta,$$

which concludes that $\mathbb{P}\left(\mathcal{E}\right) \geq 1 - \delta$.

The rest of the proof computes the regret for when $\mathcal{E}$ holds. For simplicity and without loss of generality, we assume all expert probabilities are dividable by $p_{\min}$. Recall that we follow a deterministic sampling approach and choose each arm according to its relative frequency $\frac{p(\cdot)}{p_{\min}}$ for multiple batches, where each batch loops over all active actions. Let $t_a$ be the episode in which we eliminate an arm $a$ in favour of another arm. Then, it is easy to show that

$$\forall a' \in \text{active arms by } t_a : \ p(a') \cdot t_a \le n_{t_a}(a'), \tag{20}$$

This lower bound corresponds to the case where no other arm is eliminated before eliminating $a$. Moreover, we have an upper bound for $n_{t_a}(a)$ considering the worst-case scenario in which the only remaining arms are $a$ and $a_c$, where $a_c$ is the optimal action for unobserved context $c$:

$$n_{t_a}(a) \le \frac{p(a)}{p(a) + p(a_c)} \cdot t_a. \tag{21}$$

Now, let $\text{Reg}_c(a)$ be the total regret contributed by the arm $a$ for a given context $c \sim \mathcal{C}$. We can upper bound the regret as

$$
\begin{aligned}
\text{Reg}_c(a) &= n_{t_a}(a) \left( V_c(a_c) - V_c(a) \right) \\
&\overset{(i)}{\le} 2n_{t_a}(a) \left( \sqrt{\frac{\log\left(4T^4 K/\delta\right)}{2n_{t_a}(a)}} + \sqrt{\frac{\log\left(4T^4 K/\delta\right)}{2n_{t_a}(a_c)}} \right) \\
&\le 2\frac{p(a)}{p(a) + p(a_c)} \cdot t_a \left( \sqrt{\frac{\log\left(4T^4 K/\delta\right)}{2n_{t_a}(a)}} + \sqrt{\frac{\log\left(4T^4 K/\delta\right)}{2n_{t_a}(a_c)}} \right) && \text{By (21)} \\
&= \sqrt{2\log\left(4T^4 K/\delta\right)} \cdot \frac{p(a)}{p(a) + p(a_c)} \cdot t_a \left( \sqrt{\frac{1}{n_{t_a}(a)}} + \sqrt{\frac{1}{n_{t_a}(a_c)}} \right) \\
&\le \sqrt{2\log\left(4T^4 K/\delta\right)} \cdot \frac{p(a)}{p(a) + p(a_c)} \cdot t_a \left( \sqrt{\frac{1}{t_a p(a)}} + \sqrt{\frac{1}{t_a p(a_c)}} \right) && \text{By (20)} \\
&= \sqrt{2t_a \log\left(4T^4 K/\delta\right)} \cdot \frac{p(a)}{p(a) + p(a_c)} \left( \sqrt{\frac{1}{p(a)}} + \sqrt{\frac{1}{p(a_c)}} \right) \\
&\overset{(ii)}{\le} \sqrt{2T \log\left(4T^4 K/\delta\right)} \cdot \frac{p(a)}{p(a) + p(a_c)} \left( \sqrt{\frac{1}{p(a)}} + \sqrt{\frac{1}{p(a_c)}} \right),
\end{aligned}
$$

where $(i)$ holds since the confidence intervals of arm $a$ and $a_c$ overlap at episode $t_a$ (otherwise, $a$ would have been eliminated before $t_a$), and $(ii)$ follows from the fact that $t_a \le T$.

Finally, we upper bound the Bayesian regret by taking the expectation of $\sum_{a \ne a_c} \text{Reg}_c(a)$ over $c \sim \mathcal{C}$. Note that since the expert is optimal, we have $p(a) = \mu^\star(a_c = a)$ for all $k \in \mathcal{A}$.

$$
\begin{aligned}
\text{Reg} = \mathbb{E}_{c \sim \mu^\star} \left[ \sum_{a \ne a_c} \text{Reg}_c(a) \right] &\overset{(i)}{\le} \sum_{a' \in \mathcal{A}} \mu^\star\left( a_c = a' \right) \left( \max_{c; a_c = a'} \sum_{a \ne a'} \text{Reg}_c(a) \right) \\
&= \sum_{a' \in \mathcal{A}} p(a') \left( \max_{c; a_c = a'} \sum_{a \ne a'} \text{Reg}_c(a) \right)
\end{aligned}
$$

where $(i)$ follows by partitioning $\mathcal{C}$ into $\{c \,;\, c \in \mathcal{C}, \ a_c = a'\}_{a' \in \mathcal{A}}$ and choosing the worst-case context in each partition.

From above, we have

$$\text{Reg} \leq \sum_{a' \in \mathcal{A}} p(a') \left( \max_{c; a_c = a'} \sum_{a \neq a'} \text{Reg}_c(a) \right)$$

$$\leq \sqrt{2T \log\left(4T^4 K/\delta\right)} \sum_{a' \in \mathcal{A}} \sum_{a \neq a'} \frac{p(a')p(a)}{p(a) + p(a')} \left( \sqrt{\frac{1}{p(a)}} + \sqrt{\frac{1}{p(a')}} \right)$$

$$\overset{(ii)}{\leq} \sqrt{8T \log\left(4TK/\delta\right)} \sum_{a,a' \in \mathcal{A}; a \neq a'} \frac{p(a')p(a)}{p(a) + p(a')} \left( \sqrt{\frac{1}{p(a)}} + \sqrt{\frac{1}{p(a')}} \right)$$

$$= \sqrt{8T \log\left(4TK/\delta\right)} \sum_{a,a' \in \mathcal{A}; a \neq a'} \sqrt{\frac{p(a')}{p(a) + p(a')} \cdot \frac{p(a)}{p(a) + p(a')}} \left( \sqrt{p(a)} + \sqrt{p(a')} \right)$$

where $(ii)$ holds since $4K/\delta > 1$. Replacing $\frac{p(a)}{p(a)+p(a_c)}$ with $1 - \frac{p(a')}{p(a)+p(a_c)}$ concludes the proof. $\square$

## A.5 Max-Entropy Expert Posterior for MDPs

**Proposition 6** (Max-Entropy Expert Posterior for MDPs). *Consider a contextual MDP $\mathcal{M} = (\mathcal{S}, \mathcal{A}, \mathcal{T}, R, H, \rho, \mu^\star)$. Assume the transition function $\mathcal{T}$ does not depend on the context variables. Moreover, assume the reward distribution is Gaussian with unit variance and Assumption 1 holds. Then, the log-pdf posterior function under the maximum entropy prior is given as:*

$$\forall \boldsymbol{\theta} \in \boldsymbol{\Theta}: \ \log \mu_{ME}\left(\boldsymbol{\theta} \mid \mathcal{H}_T\right) = -\sum_{t=1}^{T} \sum_{h=1}^{H} \frac{1}{2} \left( r_h^t + \max_{a' \in \mathcal{A}} \mathbb{E}_{s'}\left[Q\left(s', a'; \boldsymbol{\theta}\right)\right] - Q\left(s_h^t, a_h^t; \boldsymbol{\theta}\right) \right)^2$$

$$+ \sum_{\tau \in \mathcal{D}_E} \alpha_\tau^\star \cdot \prod_{(s,a) \in \tau} \frac{\exp\left\{\beta \cdot Q\left(s, a; \boldsymbol{\theta}\right)\right\}}{\sum_{a' \in \mathcal{A}} \exp\left\{\beta \cdot Q\left(s, a'; \boldsymbol{\theta}\right)\right\}} + \text{constant in } \boldsymbol{\theta}, \tag{22}$$

*where $\mathcal{H}_T = \left\{ \left( \left(s_h^t, a_h^t, r_h^t, s_{h+1}^t\right)_{h=1}^{H} \right)_{t=1}^{T} \right\}$ is the history of online interactions, $\mathcal{D}_E$ is the expert demonstration data, $\beta$ is the competence level of the expert in Assumption 1, and $\{\alpha_\tau^\star\}_{\tau \in \mathcal{D}_E}$ are derived from Proposition 1.*

**Remark.** We note that, in principle, the ExPerior framework allows for context-dependent transition functions. In this case, the log-pdf in (22) provides an optimistic upper bound on the true posterior log-pdf function. See Hao et al. [23] for a similar analysis. We leave the general case for future work. Note that the second term of (22) is simply the log-pdf of the max-entropy prior.

*Proof.* Since the transition function is context-independent, the likelihood of an expert trajectory $\tau_E$ can be simplified as:

$$\forall c \in \mathcal{C}: \quad m_{\tau_E}(c) = \prod_{h=1}^{H} p_E\left(a_h \mid s_h; c\right) \cdot \prod_{h=1}^{H} \mathcal{T}\left(s_{h+1} \mid s_h, a_h\right). \tag{23}$$

The second term in (23) is constant in $c$. This implies that the likelihood function $m_{\tau_E}(c)$ will depend on $c$ only through the expert policy, which itself is a function of optimal Q-functions by Assumption 1. Note that the second term in the definition of $m_{\tau_E}$ can be simply removed since we can re-weight the parameters $\boldsymbol{\alpha}$ in the optimization step (3) of Proposition 1. Hence, assuming the deep Q-network is expressive enough, without loss of generality, we can re-define the likelihood function of an expert trajectory $\tau_E = (s_1, a_1, s_2, a_2, \ldots, s_H, a_H, s_{H+1})$ as

$$\forall \boldsymbol{\theta} \in \Theta: \quad m_{\tau_E}(\boldsymbol{\theta}) = \prod_{h=1}^{H} \frac{\exp\left\{\beta \cdot Q\left(s_h, a_h; \boldsymbol{\theta}\right)\right\}}{\sum_{a' \in \mathcal{A}} \exp\left\{\beta \cdot Q\left(s_h, a'; \boldsymbol{\theta}\right)\right\}}.$$

We can now write the log-pdf of the posterior distribution of $\boldsymbol{\theta}$ given $\mathcal{H}_T$:

$$
\begin{aligned}
&\log \mu_{\mathrm{ME}} \left( \boldsymbol{\theta} \mid \mathcal{H}_T \right) \\
&\quad = \log \mathrm{P} \left( \mathcal{H}_T \mid \boldsymbol{\theta} \right) + \log \mu_{\mathrm{ME}}(\boldsymbol{\theta}) + \text{constant in } \boldsymbol{\theta} \\
&\quad = \sum_{t=1}^{L} \sum_{h=1}^{H} \log \rho \left( s_1^t \right) + \log R \left( r_h^t \mid s_h^t, a_h^t ; \boldsymbol{\theta} \right) + \log \mathcal{T} \left( s_{h+1}^t \mid s_h^t, a_h^t \right) + \log \mu_{\mathrm{ME}}(\boldsymbol{\theta}) + \text{const.} \\
&\quad = \sum_{t=1}^{L} \sum_{h=1}^{H} \log R \left( r_h^t \mid s_h^t, a_h^t ; \boldsymbol{\theta} \right) + \log \mu_{\mathrm{ME}}(\boldsymbol{\theta}) + \text{const.},
\end{aligned}
\tag{24}
$$

Now, given the Bellman equations, we can write the mean value of the reward function as

$$
\forall s \in \mathcal{S}, a \in \mathcal{A}: \quad \mathbb{E}\left[R\left(s, a ; \boldsymbol{\theta}\right)\right] = Q\left(s, a ; \boldsymbol{\theta}\right) - \max_{a' \in \mathcal{A}} \mathbb{E}_{s'}\left[Q\left(s', a' ; \boldsymbol{\theta}\right)\right]
$$

The reward distribution is Gaussian with unit variance. Therefore,

$$
\forall s \in \mathcal{S}, a \in \mathcal{A}, r \in \mathbb{R}: \quad R\left(r \mid s, a ; \boldsymbol{\theta}\right) = \mathcal{N}\left(Q\left(s, a ; \boldsymbol{\theta}\right) - \max_{a' \in \mathcal{A}} \mathbb{E}_{s'}\left[Q\left(s', a' ; \boldsymbol{\theta}\right)\right], 1\right).
\tag{25}
$$

Moreover, by Proposition 1, the log-pdf of the maximum entropy expert prior is given as

$$
\forall \boldsymbol{\theta} \in \Theta: \quad \log \mu_{\mathrm{ME}}(\boldsymbol{\theta}) = \sum_{\tau \in \mathcal{D}_{\mathrm{E}}} \alpha_\tau^\star \cdot m_\tau(\boldsymbol{\theta}) = \sum_{\tau \in \mathcal{D}_{\mathrm{E}}} \alpha_\tau^\star \cdot \prod_{(s,a) \in \tau} \frac{\exp\left\{\beta \cdot Q\left(s, a ; \boldsymbol{\theta}\right)\right\}}{\sum_{a' \in \mathcal{A}} \exp\left\{\beta \cdot Q\left(s, a' ; \boldsymbol{\theta}\right)\right\}}.
\tag{26}
$$

Combining (24) to (26), we conclude the proof. $\qquad\square$

### A.6 Ensemble Marginal Likelihood

**Proposition 3.** *Consider a contextual MDP $\mathcal{M} = (\mathcal{S}, \mathcal{A}, \mathcal{T}, R, H, \rho, \mu^\star)$. Assume the transition function $\mathcal{T}$ does not depend on the context variables and Assumption 1 holds. Then, the negative marginal log-likelihood of expert data $\mathcal{D}_E$ under the ensemble prior $\mu_{\boldsymbol{\theta}_{\mathrm{ens}}}$ is upper bounded by*

$$
-\log \mathrm{P}_E\left(\mathcal{D}_E ; \mu_{\boldsymbol{\theta}_{\mathrm{ens}}}\right) \leq \frac{1}{L} \sum_{i=1}^{L} \sum_{\tau \in \mathcal{D}_E} \sum_{(s,a) \in \tau} \log \left(\sum_{a' \in \mathcal{A}} \exp\left\{\beta \cdot Q\left(s, a' ; \boldsymbol{\theta}_{\mathrm{ens}}^i\right)\right\}\right) - \beta \cdot Q\left(s, a ; \boldsymbol{\theta}_{\mathrm{ens}}^i\right),
$$

*where $\beta$ is the competence level of the expert in Assumption 1.*

*Proof.* Recalling (1), the log-likelihood of the expert trajectories $\mathcal{D}_{\mathrm{E}}$ under $\mu_{\boldsymbol{\theta}_{\mathrm{ens}}}$ is given by

$$
\begin{aligned}
-\log \mathrm{P}_{\mathrm{E}}\left(\mathcal{D}_{\mathrm{E}} ; \mu_{\boldsymbol{\theta}_{\mathrm{ens}}}\right) &= \sum_{\tau^{(i)} \in \mathcal{D}_{\mathrm{E}}} -\log \mathbb{E}_{\boldsymbol{\theta} \sim \mu_{\boldsymbol{\theta}_{\mathrm{ens}}}} \left[\rho(s_1^{(i)}) \prod_{h=1}^{H} p_{\mathrm{E}}\left(a_h^{(i)} \mid s_h^{(i)} ; \boldsymbol{\theta}\right) \mathcal{T}\left(s_{h+1}^{(i)} \mid s_h^{(i)}, a_h^{(i)}\right)\right] \\
&= \sum_{\tau^{(i)} \in \mathcal{D}_{\mathrm{E}}} -\log \mathbb{E}_{\boldsymbol{\theta} \sim \mu_{\boldsymbol{\theta}_{\mathrm{ens}}}} \left[\prod_{h=1}^{H} p_{\mathrm{E}}\left(a_h^{(i)} \mid s_h^{(i)} ; \boldsymbol{\theta}\right)\right] + \text{constant in } \boldsymbol{\theta}_{\mathrm{ens}} \\
&\qquad\qquad\qquad\qquad\qquad\qquad\qquad\qquad\qquad\qquad\qquad (\rho, \mathcal{T} \text{ do not depend on } \boldsymbol{\theta}) \\
&= \sum_{\tau^{(i)} \in \mathcal{D}_{\mathrm{E}}} -\log \left(\frac{1}{L} \sum_{j=1}^{L} \prod_{h=1}^{H} p_{\mathrm{E}}\left(a_h^{(i)} \mid s_h^{(i)} ; \boldsymbol{\theta}_{\mathrm{ens}}^j\right)\right) \qquad (\text{By Definition of } \mu_{\boldsymbol{\theta}_{\mathrm{ens}}}) \\
&\leq \sum_{\tau^{(i)} \in \mathcal{D}_{\mathrm{E}}} \frac{1}{L} \sum_{j=1}^{L} \sum_{h=1}^{H} -\log p_{\mathrm{E}}\left(a_h^{(i)} \mid s_h^{(i)} ; \boldsymbol{\theta}_{\mathrm{ens}}^j\right) \qquad \text{By Jensen's inequality} \\
&= \frac{1}{L} \sum_{i=1}^{L} \sum_{\tau \in \mathcal{D}_{\mathrm{E}}} \sum_{(s,a) \in \tau} \left[\log \left(\sum_{a' \in \mathcal{A}} \exp\left\{\beta \cdot Q\left(s, a' ; \boldsymbol{\theta}_{\mathrm{ens}}^i\right)\right\}\right) - \beta \cdot Q\left(s, a ; \boldsymbol{\theta}_{\mathrm{ens}}^i\right)\right] \\
&\qquad\qquad\qquad\qquad\qquad\qquad\qquad\qquad\qquad\qquad\qquad\qquad\qquad \text{By Assumption 1}
\end{aligned}
$$

$\qquad\square$

