# OpenReview forum: "Sequential Decision Making with Expert Demonstrations under Unobserved Heterogeneity"
_NeurIPS.cc/2024/Conference — NeurIPS 2024 poster_

### Official Review · Reviewer_FGqG · 2024-06-16

**Soundness:** 3
**Presentation:** 3
**Contribution:** 2
**Rating:** 6
**Confidence:** 4

**Summary:**

Imitation learning in a setting where experts have contextual information not available to the imitator. Imitator sees state-action (no reward) trajectories for the expert data. The paper proposes to use the expert data as prior and then do meta-RL to learn a policy.

**Strengths:**

- The problem setting considered is important and there isn't too much prior work in it.
- The proposed method seems sensible.
- The paper provides theoretical and empirical justification for the proposed method.

**Weaknesses:**

- The empirical evaluation is limited to simple domains and it is not discussed why that is the case. It would be good to include an explanation on why this was not scaled up to some more complex environments.
- Comparison to ExPLORe: if there is a way to use the same actor-critic algorithm for the proposed method and the comparison that would strengthen the paper. As the text says, now it is not clear whether the difference comes from the online RL part or from the difference in how they use the expert data.
- ADVISOR [1] targets the same problem setting. Differences to that algorithm should be discussed. And maybe a pointer should be given on why their method is demonstrated in high-dimensional domains and this isn't. Not saying that this has to be evaluated in high-dimensional domains, but that it is not immediately clear why this isn't.
- Figure 4
    - Explain the pane meanings explicitly in the caption.
    - Explain the axes explicitly in the caption. I.e., are these learning curves?
    - How come the naive baseline does worse in the low entropy setting than in the high entropy one?

[1] Luca Weihs, Unnat Jain, Iou-Jen Liu, Jordi Salvador, Svetlana Lazebnik, Aniruddha Kembhavi, and Alex Schwing. Bridging the imitation gap by adaptive insubordination. Advances in Neural Information Processing Systems, 34:19134–19146, 2021.

**Questions:**

- Why is human-in-the-loop highlighted as a limitation?
- Can you give a more fleshed out example of how the problem of unobserved contextual information might appear in a realistic problem?
- Can you lay out an idea on how to scale the method?

**Limitations:**

Limitations are discussed but some of the limitations highlighted seem a little strange. See questions.

---

> ### Author Rebuttal · Authors · 2024-08-05
>
> Thank you for your thoughtful and in-depth feedback. We are encouraged that you find the problem setting important, and the proposed method sensible with theoretical and empirical justification. We address your questions below.
>
> > The empirical evaluation is limited to simple domains and it is not discussed why that is the case...
>
> Please see the General Response for our detailed answer.
>
> > Comparison to ExPLORe: if there is a way to use the same actor-critic algorithm for the proposed method ...
>
> Thank you for mentioning this interesting point. As we have discussed in the paper, we agree with you that some of the results might be due to the difference between the base actor-critic method in ExPLORe and the bootstrapped DQN method used in our approach. However, note that in Table 1, on the left side (Fixed # Hazard = 9), under $\beta = 0.1$, the ExPLORe method achieves near-optimal results while our approaches suffer large regrets. This particular example shows that, at least in the Frozen Lake environment, the baseline actor-critic method is superior to the baseline bootstrapped DQN since the expert data is relatively uninformative for $\beta = 0.1$, i.e., the taken actions are almost random. As we increase the value of $\beta$ and the expert data gets more informative, our proposed approaches outperform the naive bootstrapped DQN, showing that the main performance comes from the proposed method of learning the priors and not the base bootstrapped DQN algorithm. We leave the variant of ExPerior that relies on actor-critic methods to future work.
>
> > ADVISOR [1] targets the same problem setting. Differences to that algorithm should be discussed ...
>
> The main difference to ADVISOR is that they assume having access to the expert policy $\pi^{\text{teach}}$, i.e., they have a human-in-the-loop setting during training and can query the expert for new states. On the other hand, our approach only assumes a finite offline data from the expert trajectories. In other words, there is no expert demonstration for new states that the learner can encounter. We will elaborate on the difference in the revised version. Please see our General Response for the high-dimensional domains.
>
> > Figure 4: (1) Explain the pane meanings explicitly in the caption. (2) Explain the axes explicitly in the caption. I.e., are these learning curves?
>
> Thank you for the suggestions. We will include the meaning of each panel (low-entropy to high-entropy setting) in the caption. The y-axis refers to the expected reward for some fixed number of evaluation episodes at each training episode $t \leq T$. We will include that in the caption as well.
>
> > Figure 4: How come the naive baseline does worse in the low entropy setting than in the high entropy one?
>
> This is an astute observation. The reason is that as the entropy of the tasks gets higher, i.e., the goal location gets uniformly distributed over all the columns, the task will intrinsically get easier. Since the agent always starts from the top-left corner, it is much harder for it to explore and reach the bottom-right corner compared to bottom-middle or bottom-left cells. This is why, on average, the naive baseline gets higher rewards in the high-entropy case compared to the low-entropy setting (the goal column always being in the right corner).
>
>
> > Q. Why is human-in-the-loop highlighted as a limitation?
>
> By human-in-the-loop, we mean the requirement of the algorithm to query experts during training or inference. Our approach, on the other hand, only requires an offline dataset of expert demonstrations, which is a strictly simpler requirement. It might be difficult to access an online expert (e.g., a clinical doctor) during training. This is what we mean by a limitation. Having said that, there are benefits in human-centric AI and looping humans in decision-making that we haven’t discussed in our paper. We will elaborate on this point in the paper’s Conclusion.
>
> > Q. Can you give a more fleshed out example of how the problem of unobserved contextual information might appear in a realistic problem?
>
> In clinical applications, the objective descriptions of a patient are often documented through electronic health records (EHR). However, the subjective opinion of a doctor after having conversations with the patient and observing their visual clues or other information, such as their lifestyle, might lead to potentially different diagnoses. Even by adding clinical notes to the EHR data, it is fundamentally challenging to capture all the observations made by the doctors that influence their decisions. Another example is large language models, which are trained on datasets that do not capture contextual information. For example, suppose we ask an expert about the location of a historical building. If the expert remembers that as part of her internal knowledge, she will answer immediately. Training LLMs on such datasets might result in hallucination as the model does not capture the contextual information – When asked about the location of another historical building, the model might think like an expert and generate some response based on their internal knowledge, which can be wrong. On the other hand, what you might expect from such a model is to search on the internet to respond. Please See [1] and [2] (Figure 7) for more detail.
>
> > Q. Can you lay out an idea on how to scale the method?
>
> Please see the General Response for our detailed answer.
>
> We hope our response addressed your comments. If you have any additional questions that could help you improve your evaluation of the work, please feel free to let us know.
>
> ===================================
>
> [1] Johnson, Daniel D., et al. "Experts Don't Cheat: Learning What You Don't Know By Predicting Pairs." arXiv preprint arXiv:2402.08733 (2024).
>
> [2] Wang, Kuan, et al. "Adapting LLM Agents with Universal Feedback in Communication." ICML 2024 Workshop on Foundation Models in the Wild.

---

> > ### Comment · Reviewer_FGqG · 2024-08-11
> >
> > Thank you for the response. Most of my concerns have been addressed. I'm raising my score.

---

### Official Review · Reviewer_2ZF5 · 2024-07-08

**Soundness:** 3
**Presentation:** 2
**Contribution:** 2
**Rating:** 5
**Confidence:** 2

**Summary:**

The paper proposes a 2-stage learning method for sequential decision making, wherein step 1 involves leaning a prior from expert demonstrations and step 2 is online RL where that learnt prior is used. The main problem being addressed is that of heterogeneity of data and/or contexts provided by the expert and those encountered by the online learning agent.

**Strengths:**

- Relevant problem: Finding useful, scalable and efficient ways to leverage offline expert demonstrations for online decision making is of key interest to the community.

- The proposed method combines imitation learning with online model-based RL, paying attention to the issue of heterogeneity between the available contexts that gave rise to the offline data, that might be unavailable during the online RL step.

- The problem is well-motivated and the approach is easy to understand conceptually; contributions are clearly outlined.

- Presentation (writing, figures etc) is mostly clear, with some minor comments included in the below section.

**Weaknesses:**

Clarity
- Notation: even with the notation paragraph, I find it quite difficult to follow. E.g. I think some `t` superscripts are missing (the definition of the q-function); should $\mathcal{C}$ be part of the definition of the MDP?
- Terminology: I'm a little confused with the terminology "parametric" vs "non-parametric". Unless I'm missing something, the latter is still parametric as you are optimising over a family of distributions?
- Typo: in figure 1, annotation below "Step 3" is repeated, I guess one should be $\mu_{\theta}*$.
- Figure 1: clarity can be improved if you write down what is being learnt and what is being updated.
- I found the algorithm box quite useful, will be great to have it in the main paper (e.g. next to Figure 1).

Max-entropy prior:
- Clearly, if $\mu_0  \in \mathcal{P}(\epsilon)$, then we get a trivial solution, $\mu_{ME}=\mu_0$ (usually Uniform), so do you need an assumption on ${P}(\epsilon)$ to get a non-trivial solution?
- If $\mathcal{C}$ is unbounded, then you can't really define a uniform distribution on it, how do you deal with that?
- Can you handle improper priors?

**Questions:**

In addition to those raised in the Limitations section:

1. Do you have an intuition on how badly "misspecified" the learnt expert prior can get until learning for the online agent becomes very difficult? Is there a way to detect a poorly specified prior, and potentially reverting to an alternative (e.g. uninformative) after a few steps online?
2. For the "parametric" ExPrior - why score gradients? Have you thought about or tried continuous relaxation when the problem is discrete?
3. On line 112 "Note that since the task variable is unobservable, the learner's policy will not depend on it" - does it depend implicitly though, e.g. can the learner infer and integrate over that inferred task distribution?

**Limitations:**

There are no specific negative societal impact that need to be discussed.

---

> ### Author Rebuttal · Authors · 2024-08-05
>
> Thank you for your insightful feedback. We are pleased that you consider the paper well-motivated, relevant, and with a conceptually easy-to-understand approach. We have addressed your questions below.
>
> > Notation: even with the notation paragraph, I find it quite difficult to follow...?
>
> We assume the states are partitioned by the horizon $[H]$, i.e., each state $s \in \mathcal{S}$ is only reachable in one specific horizon $h \in [H]$. This can be achieved by re-defining the state space as $\mathcal{S} \times H$. This way, none of the value functions or Q-functions need to depend on the horizon. The superscript $t$, however, corresponds to the episode number and not the horizon. The value/Q-functions are defined independently of the episode. The only variable that gets updated over episodes is the learned policy $\pi^t$. Also, we agree that $\mathcal{C}$ should be part of the definition of the MDP. We will include these comments in the notation page of the revised manuscript, as there will be more space (10 pages instead of 9).
>
> > Terminology: I'm a little confused with the terminology "parametric" vs "non-parametric"...
>
> Looking at the max-entropy expert prior in Proposition 1 (line 189), you can see that.
> $$\mu\_{\text{ME}}(c) = \lim\_{k \to \infty} \frac{\exp \left( \mathbf{m}(c)^\top \boldsymbol{\alpha}_{k} \right)}{\mathbb{E}\_{c'\sim \mu\_{0}} \left[ \exp \left( \mathbf{m}(c')^\top \boldsymbol{\alpha}\_{k} \right) \right]}$$
> where each $\boldsymbol{\alpha}\_k$ is an $N$-dimensional parameter with $N$ being the number of demonstrations. In other words, the length of parameters required to model the max-entropy prior increases by the number of samples. This is what we mean by a "non-parametric" approach, in contrast to "parametric" models with a fixed set of learnable parameters. Moreover, we do not assume any specific form for the family of distributions in $\mathcal{P}(\epsilon)$, except belonging to $L^1(\mathcal{C}, \mu\_0)$ defined in lines 577-583.
>
> > Typo: in figure 1, annotation below "Step 3" is repeated, I guess one should be \mu_{\theta^\star}.
>
> This is a mistake, and we will fix it in the revised version. Thank you for pointing this out.
>
> > Figure 1: clarity can be improved if you write down what is being learnt and what is being updated.
>
> $\mu\_{\theta^\star}$, $\mu\_{\text{ME}}$ are learned from data. The posterior distributions  $\mu\_{\theta^\star}(\cdot | \text{history})$, $\mu\_{\text{ME}}(\cdot | \text{history})$ are being updated, and the policy $\pi\_c(\cdot | s)$ is learned/calculated given the sampled Q-functions. We will differentiate between those with different colours in Figure 1.
>
> > I found the algorithm box quite useful, will be great to have it in the main paper (e.g. next to Figure 1).
>
> We moved algorithm boxes 1 and 2 to the appendix solely for the lack of space. If the paper gets accepted, we will include them in the main paper. (The camera-ready version allows for one more page.)
>
>
> > Clearly, if $\mu\_0 \in \mathcal{P}(\epsilon)$, then we get a trivial solution, $\mu\_{\text{ME}} = \mu\_0$ (usually Uniform) ...?
>
> Your assessment is correct. If the uniform distribution is already in the set of feasible prior functions $\mathcal{P}(\epsilon)$, then  $\mu\_{\text{ME}} = \mu\_0$. However, this is not a trivial solution since the uniform distribution will maximize the marginal likelihood of the data and is a plausible distribution. This scenario can happen if the effect of unobserved confounding is extreme, i.e., the unobserved factors can change the optimal policies arbitrarily and provide no information on the task distribution. For example, consider a multi-armed bandit setting with $K$ actions, where there are $K$ task types, each with a different arm as the optimal action. The expert demonstration data will contain each of the arms with equal probability. In other words, the expert data provides no information, and the best thing we can do is to have a non-informative prior like the uniform prior distribution.
>
>
> > If $\mathcal{C}$ is unbounded, then you can't really define a uniform distribution on it ...? Can you handle improper priors?
>
> Note that our result does not force $\mu\_0$ to be a uniform distribution or $\mathcal{C}$ to be bounded. It only requires $\mathcal{C}$ to be a measurable set with an existing measure $\mu\_0$. In the case of unbounded $\mathcal{C}$, one can choose $\mu\_0$ to be a Gaussian distribution. However, our current results, like Lemma 5 on page 14, require the priors to be in $L^1\left(\mathcal{C}, \mu\_0\right)$, i.e., $\mathcal{C}$ should have a finite measure under the feasible priors. For that reason, the current theoretical results do not handle improper priors.
>
> > Q. Do you have an intuition on how badly "misspecified" the learnt expert prior can get ...?
>
> This is the exact reason we have provided the non-parametric max-entropy approach. Table 3 (Page 22) shows that misspecified priors result in large regrets. In cases where the practitioner is not quite sure about the parametric form of the prior, we suggest using the non-parametric max-entropy approach.
>
> > Q. For the "parametric" ExPrior - why score gradients? Have you thought about or tried continuous relaxation when the problem is discrete?
>
> In the parametric ExPerior, we do not assume the parameter set $\boldsymbol{\Theta}$ is necessarily continuous, as long as we can solve the optimization in line 175 (potentially without gradient-based methods).
>
> > Q. On line 112 "Note that since the task variable is unobservable ...
>
> We meant explicit dependence here. The learner policy only depends on the history of interactions and the current state. As you mentioned, one can use the history to infer/integrate the task distribution. We will clarify more on this.
>
> We hope our comment addresses your questions. Please let us know if we can answer any further questions that might improve your assessment of the work.

---

### Official Review · Reviewer_hfW9 · 2024-07-11

**Soundness:** 3
**Presentation:** 3
**Contribution:** 3
**Rating:** 6
**Confidence:** 3

**Summary:**

This paper attempt to leverage offline demonstrations to speed up online learning under unobserved heterogeneity, and unlike zero-shot meta reinforcement learning, the proposed ExPerior does not require the task labels (reward labels). ExPerior utilizes expert data to establish an informative prior distribution for online exploration. The experimental results on multi-armed bandits and MDPs showcase the superiority against current baselines.

**Strengths:**

The setting proposed in this paper is realistic and significant. To solve the problem that the learner faces uncertainty and variability in task parameters, the paper proposes two approaches, parametric and non-parametric, and provides sufficient theoretical basis. Experiments under multi-armed bandits and MDPs also prove its effectiveness.

**Weaknesses:**

1.	Just as stated at the end of the paper, the experiments conducted in the paper are limited, and the experiments under MDPs even made the assumption that the transition functions are invariant to the task variables, which makes the experimental environment overly simple. In more difficult experimental benchmarks such as MuJoCo, it is questionable whether the proposed method can show the expected effect.
2.	Due to the deficiency mentioned in the first point just now, the paper also cannot further compare with modern offline meta RL algorithms (such as PEARL, FOCAL, CORRO) under complex MDPs, making the baselines seem a bit scarce.
3.	The introduction of related concepts in the paper is slightly insufficient. Besides, there is a clerical error in the upper right part of Fig. 1: the first \mu_{\rm{ME}} should be \mu_{\theta^*}.

**Questions:**

1.	What does "Heterogeneity" mentioned in the paper represent? Does it refer to the differences between the transition functions and reward functions in MDPs? The paper should explain this.
2.	The paper mentions that the off-policy meta RL method PEARL requires task labels. Does the task label here refer to the reward label? In fact, the trajectories required by PEARL only contain (s, a, r, s') information.

**Limitations:**

A/N

---

> ### Author Rebuttal · Authors · 2024-08-05
>
> Thank you for your thoughtful feedback. We are encouraged that you find the paper to have a realistic and significant setting and sufficient theoretical and experimental results. We address your questions below.
>
>
> > Just as stated at the end of the paper, the experiments conducted in the paper are limited, and the experiments under MDPs even made the assumption that the transition functions are invariant to the task variables, which makes the experimental environment overly simple. In more difficult experimental benchmarks such as MuJoCo, it is questionable whether the proposed method can show the expected effect.
>
> We agree that the current experiments do not consider complex high-dimensional settings. Please see the General Response we provided for our detailed plan to scale our approach and include such experiments in the revised version.
>
> > Due to the deficiency mentioned in the first point just now, the paper also cannot further compare with modern offline meta RL algorithms (such as PEARL, FOCAL, CORRO) under complex MDPs, making the baselines seem a bit scarce.
>
> Thank you for mentioning the works FOCAL [1] and CORRO [2]. We will cite these in the related work section of the revised paper. However, independent of the complexity of the MDPs, the mentioned baselines are not immediately applicable for our problem setup. PEARL, FOCAL, and CORRO all require reward signals to be available in the offline data. Moreover, FOCAL and CORRO assume the offline data is diverse enough to learn the optimal policies solely based on the offline data and do not continue the learning during the online phase. On the other hand, our setting only assumes offline expert demonstration data (with no reward) and requires using the reward signals during the online phase. This is why we did not include those baselines in our experiments, rather than the complexity of the MDPs.
>
> > The introduction of related concepts in the paper is slightly insufficient. Besides, there is a clerical error in the upper right part of Fig. 1: the first \mu_{\rm{ME}} should be \mu_{\theta^*}.
>
> Could you please elaborate on which parts of the introduced concepts are slightly insufficient so we can explain them in more detail? Also, thank you for finding the typo in $\mu_{\rm{ME}}$. We will fix it in the updated version.
>
> > Q: What does "Heterogeneity" mentioned in the paper represent? Does it refer to the differences between the transition functions and reward functions in MDPs? The paper should explain this.
>
> Yes, your understanding is accurate. We will elaborate more on what we mean by "Heterogeneity" in the introduction.
>
> > Q: The paper mentions that the off-policy meta RL method PEARL requires task labels. Does the task label here refer to the reward label? In fact, the trajectories required by PEARL only contain (s, a, r, s') information.
>
> Thank you for pointing this out. This confusion is probably due to the paper's phrasing. Quoting the paper (lines 85-86): "Similarly, Zhou et al. [40] and Rakelly et al. [41] require the task label and reward labels.". This should have instead been, "Similarly, Zhou et al. [40] and Rakelly et al. [41] require the task label and reward labels, *respectively*." PEARL requires offline reward information, which is not accessible in our setting. We will fix this confusion in the revised version.
>
> We hope our response addressed your comments. Should you have any additional questions that could assist in improving your evaluation of the work, please feel free to let us know.
>
> ===================================
>
> [1] Li, Lanqing, Rui Yang, and Dijun Luo. "Focal: Efficient fully-offline meta-reinforcement learning via distance metric learning and behavior regularization." arXiv preprint arXiv:2010.01112 (2020).
>
> [2] Yuan, Haoqi, and Zongqing Lu. "Robust task representations for offline meta-reinforcement learning via contrastive learning." International Conference on Machine Learning. PMLR, 2022.

---

> > ### Comment · Reviewer_hfW9 · 2024-08-12
> >
> > Thank you for your response, I maintain my original score.

---

### Official Review · Reviewer_11BA · 2024-07-11

**Soundness:** 3
**Presentation:** 3
**Contribution:** 2
**Rating:** 5
**Confidence:** 3

**Summary:**

This paper studies online transfer learning in an unknown Markov decision process. The learner has access to demonstration trajectories generated by an expert, who has access to an independent context variable. The expert can observe the values of the latent context to make a decision, but such information is unobserved to the learner. The authors propose a learning strategy that allows the learner to accelerate its online learning process by leveraging the confounded expert's trajectories. More specifically, the expert selects values of the action following a specific parametric family of sofmax policy. Using this parametric information, the learner can extrapolate an informative prior about the underlying system dynamics (i.e., transition function and reward function) from the confounded trajectories and utilize the extrapolated prior to improve future learning. The authors then derive a Bayesian regret bound for the proposed method. Simulation results support the proposed approach.

**Strengths:**

- The paper is well-organized and clearly written. Simulations are comprehensive, supporting the proposed approach.
- Theoretica regret analysis is provided. I have not read through all the proofs for Theorem 2. But the result seems reasonable.

**Weaknesses:**

- The proposed method requires the expert to follow a specific form of policy to generate the demonstration data. This parametric restriction is rather strong and may not necessarily hold in many applications.
- It is unclear to see from Theorem 2 under which condition the expert's demonstration could accelerate the online learning process. It would be appreciated if the authors could further elaborate on this point.
- Some related references are missing. Extrapolating knowledge from confounded data is one of the main problems in causal inference. There is a growing line of work studying learning from confounded demonstration data in canonical RL tasks, including:

1. Kallus, Nathan, and Angela Zhou. "Confounding-robust policy improvement." Advances in neural information processing systems 31 (2018).
2. Zhang, Junzhe, and Elias Bareinboim. "Near-optimal reinforcement learning in dynamic treatment regimes." Advances in Neural Information Processing Systems 32 (2019).

**Questions:**

- Could the regret bound in Theorem 2 outperform standard bandit regret when the expert's demonstration is benign? Could the authors elaborate on this?

**Limitations:**

The authors have adequately addressed the limitations of the paper.

---

> ### Author Rebuttal · Authors · 2024-08-05
>
> Thank you so much for your helpful comments. We are pleased that you find the paper clearly written, well-organized, and with comprehensive simulations. We address your questions in the following.
>
> > The proposed method requires the expert to follow a specific form of policy to generate the demonstration data. This parametric restriction is rather strong and may not necessarily hold in many applications.
>
> We agree with you that Assumption 1, about the expert policy being noisily rational, may not hold in practice. Appendix C.1, Table 2 on Page 22 considers misspecified expert models. In particular, we looked into “optimal” experts, “noisily rational” experts with misspecified values of $\beta$, and a new type of experts we called “random-optimal,” who take completely random actions with some fixed probability and optimal actions otherwise. In all instances, our methods, which rely on the noisily rational experts, achieve near-optimal results. This observation justifies the use of Assumption 1, even in instances where it does not necessarily hold.  We would appreciate it if the reviewer has any suggestions for including other types of misspecification in Table 2.
>
> > It is unclear to see from Theorem 2 under which condition the expert's demonstration could accelerate the online learning process. It would be appreciated if the authors could further elaborate on this point.
>
> As discussed in lines 302-312, as well as Figure 3 (b), the extra term in the regret bound of Theorem 2, i.e.,
> $$\sum\_{a, a’ \in \mathcal{A}, a \neq a’} \sqrt{\frac{p(a)}{p(a) + p(a’)} \left(1 - \frac{p(a)}{p(a) + p(a’)}\right)} \left[\sqrt{p(a)} + \sqrt{p(a’)}\right]$$
> closely resembles the variance (or the entropy in the case of $K = 2$) of the optimal actions under the distribution of unobserved factors. In other words, **if the variance of the optimal actions is small, i.e., the effect of unobserved factors is negligible, the regret will be close to zero**. We will elaborate more on this point in the revised version of the paper.
>
> > Some related references are missing. Extrapolating knowledge from confounded data is one of the main problems in causal inference. There is a growing line of work studying learning from confounded demonstration data in canonical RL tasks ...
>
> Thank you for providing the missing related references. We will include those papers in the revised manuscript. We emphasize that the first work, Kallus et al. (2018), only considers the offline setting, while the latter assumes the availability of rewards in the offline observational data and does not assume the demonstrations are generated by experts. Our work, on the other hand, integrates **offline expert** demonstration data with **online** reinforcement learning.
>
> > Q: Could the regret bound in Theorem 2 outperform standard bandit regret when the expert's demonstration is benign? Could the authors elaborate on this?
>
> Yes, as discussed above and in lines 302-312 of the manuscript, the standard bandit regret is bounded by $\tilde{\mathcal{O}}\left(KT\right)$ for $K$ number of actions and $T$ episodes. The bound in Theorem 2, however, is
> $$\tilde{\mathcal{O}}\left(\sqrt{T} \cdot \underbrace{\sum\_{a, a’ \in \mathcal{A}, a \neq a’} \sqrt{\frac{p(a)}{p(a) + p(a’)} \left(1 - \frac{p(a)}{p(a) + p(a’)}\right)} \left[\sqrt{p(a)} + \sqrt{p(a’)}\right]}\_{\text{term} I}\right)$$
> When the expert’s demonstration is benign, e.g., the variance of the optimal actions under unobserved confounding is small, the second term (term I) will be smaller than $\sqrt{K}$. As an example, consider a $K$-armed case, where a given fixed arm $a_1$ is optimal regardless of the unobserved heterogeneity. Then, for all $a \neq a_1$, we have $p(a) = 0$, resulting in term I to be zero. This is a case when the expert data is extremely informative and hence gives us zero regret.
>
> We hope our comment addresses your questions. Please let us know if we can answer any further questions that might improve your assessment of the work.
>
> *1. The big-O notation $\tilde{\mathcal{O}}$ in this comment ignores the log terms.*

---

> > ### Comment · Reviewer_11BA · 2024-08-12
> >
> > I appreciate the authors' response. They have answered my questions. Theorem 2 states that the proposed algorithm is able to achieve performance improvement when the expert always picks the optimal action in demonstrations. This is somewhat expected since, in this case, the latent is not effective, and the demonstration data could be directly transferred. On the other hand, this is quite an extreme condition that might be hard to reach in practical conditions. I will keep my current score.

---

### Author Rebuttal · Authors · 2024-08-05

We thank all the reviewers for their helpful and thorough feedback. We are happy that they found our work clearly written, well-organized, and with comprehensive simulations (Reviewer ${\color{red} \text{11BA}}$), significant with a realistic setting and sufficient theoretical basis (${\color{blue} \text{hfW9}}$), well-motivated, easy to understand, with clearly-outlined contributions (${\color{orange} \text{2ZF5}}$), and sensible with an important setting and theoretical and empirical justification (${\color{ForestGreen} \text{FGqG}}$). We discuss the common concern on the simplified experiments and ongoing plans to scale our method, ExPerior, to high-dimensional settings. We answer each reviewer's specific questions and concerns separately.

**Limitated Empirical Evaluation (${\color{blue} \text{hfW9}}$, ${\color{ForestGreen} \text{FGqG}}$)**

1. First, we emphasize that we have run our proposed methods on bandits, MDPs and POMDPS where, in practically all instances, our proposed algorithm outperforms the baselines. To elaborate, within each setting we study:
  a. Multi-armed bandits (with various prior functions, misspecified expert models, misspecified prior functions, and under both exact posterior sampling and stochastic gradient Langevin dynamics),
  b. Markov Decision Processes (MDPs) in the Deep Sea and Frozen Lake environments, and
  c. Partially Observable MDPs (POMDPs) in the Frozen Lake environment.
To our knowledge, most papers in the literature only consider a subset of those settings (bandits, MDPs, or POMDPs) for their experiments, and we view the breadth of applicability of our approach to be a considerable strength of ExPerior.

2. The environments used in our experiments, i.e., *Deep Sea*, *Frozen Lake*, or similar grid environments, are standard in the literature, particularly for imitation learning from experts under unobserved confounding [1,2,3,4].

We are currently working on expanding the framework to study its performance on more high-dimensional problems and/or more complex environments such as MuJoCo. Therefore, we are working on providing such experiments in the revised version.

**Scaling the Method (${\color{ForestGreen} \text{FGqG}}$)**

ExPerior, derives a prior distribution over Q-functions from expert demonstrations and initializes the Q-networks by sampled parameters from such prior. For high-dimensional settings, the Q-networks will have a large number of parameters; to handle this, we intend to:

1. Pre-train a representation network using self-supervised learning on the offline expert data, e.g., by pre-training a convolutional module to capture the meaningful state representation from images of the Frozen Lake environment (instead of using the grid representation) or the MuJoCo environment.
2. Construct the Q-network by freezing the pre-trained module and adding a learnable linear layer on top of it so it will capture different Q-functions for different values of unobserved factors. This enables us to learn and sample from a prior distribution defined over a simple linear layer instead of the entire Q-network.


===================================

[1] Warrington, Andrew, et al. "Robust asymmetric learning in pomdps." International Conference on Machine Learning. PMLR, 2021.

[2] Shenfeld, Idan, et al. "Tgrl: An algorithm for teacher guided reinforcement learning." International Conference on Machine Learning. PMLR, 2023.

[3] Hao, Botao, et al. "Bridging imitation and online reinforcement learning: An optimistic tale." arXiv preprint arXiv:2303.11369 (2023).

[4] Johnson, Daniel D., et al. "Experts Don't Cheat: Learning What You Don't Know By Predicting Pairs." arXiv preprint arXiv:2402.08733 (2024).

---

### Decision · Program_Chairs · 2024-09-25

**Decision:**

Accept (poster)

**Comment:**

The paper studies sequential decision-making in the presence of expert demonstrations based on unobserved contextual information. The reviewers have identified some valuable aspects of the paper:

- Well-motivated problem
- Good presentation

However, they have also noted several concerns:

- Assumption 1 may be very limiting
- Limited experimental evaluation in terms of environments and a lack of comparison with relevant baselines
- The conditions under which performance improvement is possible (as outlined in Theorem 2) are quite extreme (the expert always plays the optimal action).